# PRISM: Performer RS-IMLE for Single-pass Multisensory Imitation Learning

## Abstract

Robotic imitation learning typically requires models that capture multimodal action distributions while operating in real-time control rates and accommodating multiple sensing modalities. Although recent generative approaches such as diffusion models, flow matching, and Implicit Maximum Likelihood Estimation (IMLE) have achieved promising results in this domain, they satisfy only a subset of these requirements. To satisfy these requirements, we introduce PRISM, based on a batch-global rejection-sampling variant of IMLE. PRISM is a single-pass policy that couples a temporal multisensory encoder (e.g, RGB, Depth, tactile, audio, proprioception) with a linear-attention generator using a Performer architecture. We validate on MetaWorld, CALVIN, Robomimic, and a real hardware suite using a Unitree Go2 with a 7-DoF arm, wrist and shoulder RGB, tactile, audio, and proprioception sensors. PRISM matches or outperforms diffusion, flow-matching, and prior IMLE policies in terms of task success rates, robustness, and sample efficiency. In CALVIN with 10% of the data, PRISM improves the success rate by $\sim 10\%$ over IMLE, $\sim 20\%$ over flow matching, and $\sim 25\%$ over diffusion, while reducing the jerk by about $20\times$. On MetaWorld, PRISM is 5-12% on Hard/Very-Hard splits over diffusion and flow baselines. Real-world loco-manipulation shows 10–25% higher success and maintains faster inference compared to diffusion policy. These results position PRISM as a fast, accurate, and multisensory imitation policy that retains multimodal action coverage without iterative sampling.

## 1 Introduction

Imitation Learning (IL) (Atkeson & Schaal, 1997; Argall et al., 2009; Rahmatizadeh et al., 2018; Avigal et al., 2022; Yu et al., 2025; Bhaskar et al., 2024) has become a powerful framework for acquiring complex visuomotor policies directly from demonstrations. IL is particularly well-suited for real-world robotic applications where safety and precision are paramount. These advantages are magnified in settings involving *multisensory observations*, such as RGB images, depth, tactile feedback, and proprioception, which are difficult to simulate accurately and often only partially observable in practice. An ideal imitation learning policy should therefore meet three critical criteria: (i) operate at **real-time inference rates** for closed-loop physical control; (ii) represent the **multimodal distribution** of expert behavior (e.g., multiple valid grasps or approach paths); and (iii) perform robustly under **partial sensory input**, by effectively fusing multiple sensory streams.

Recent generative policies address subsets of this challenge, but rarely all. Diffusion models (Chi et al., 2023; Ze et al., 2024; Song et al., 2022) have shown impressive ability to capture complex, multimodal action distributions, but their reliance on iterative denoising (often $\sim$10–100 steps per inference) severely limits real-time deployment (Lu et al., 2024; Prasad et al., 2024). Flow-based methods (Hu et al., 2024; Funk et al., 2024; Zhang et al., 2025; Yang et al., 2024) reduce sampling steps through continuous-time integration, but may still struggle with multimodal fidelity (Rouxel et al., 2024). Moreover, these approaches are computationally expensive during inference due to iterative sampling and often require large datasets to train effectively. These trade-offs highlight the need for approaches that are both expressive and efficient.

Recently a new model, called IMLE Policy, was proposed for conditional imitation learning (Rana et al., 2025) based on Implicit Maximum Likelihood Estimation (IMLE) (Li & Malik, 2018). IMLE trains by minimizing the distance from each expert datapoint to its nearest generated sample, ensuring full coverage of the expert distribution and enabling **single-pass inference**, generating multiple

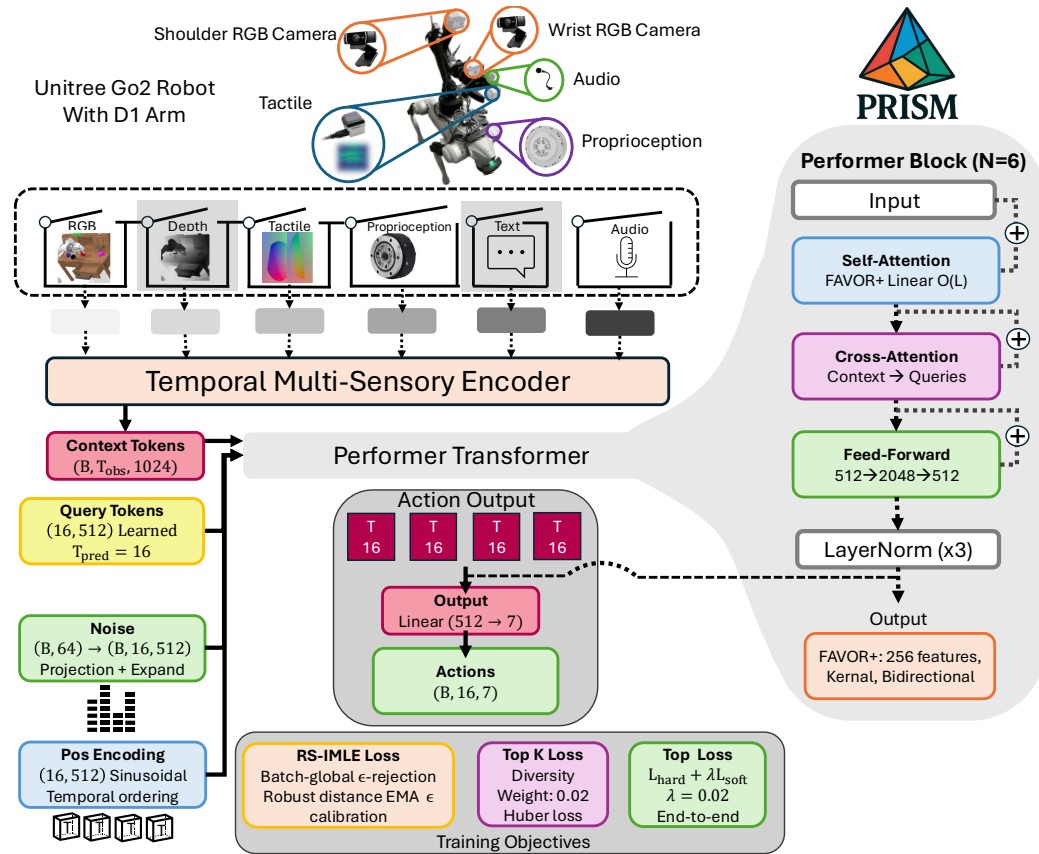

Figure 1: **Overview of Performer RS–IMLE for Multisensory Imitation (PRISM).** Per-timestep features from the available sensors are fused into temporal context tokens. Across all benchmarks we include one or more *RGB cameras*; *depth camera, tactile, proprioception, text tokens, and audio* are used when present in a dataset. A *bidirectional* FAVOR$^+$ (Performer) generator with learned query tokens outputs a full sequence of actions in a single pass. Training uses a *batch-global* RS-IMLE objective (robust Charbonnier distance, $\varepsilon$-rejection with EMA calibration, optional small coverage term) to preserve action *multimodality* without iterative sampling. We train *separate models per benchmark* (MetaWorld, CALVIN, Robomimic, real hardware) using only the modalities available in that benchmark (see Fig. 2 for list of modalities used per dataset type); unused sensors are shown in gray. At inference we sample *k* latent-conditioned trajectories in one batched pass and select one for receding-horizon control for hardware experiment.

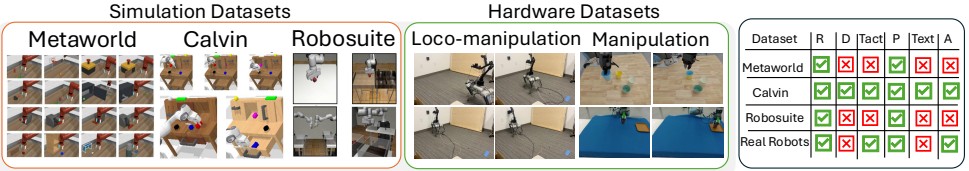

Figure 2: **Benchmark Datasets.** On the right, we show the sensor modalities used per dataset. *R* is for RGB camera, *D* is for depth camera, *Tact* is for tactile, *P* is for proprioception, *Text* is for text tokens, and *A* is for audio.

plausible actions in parallel and selecting one without iterative refinement. IMLE policy uses Rejection Sampling IMLE (RS-IMLE) (Vashist et al., 2024) which further improves candidate quality by discarding those that are too similar to ground truth, promoting diversity in training-time sampling.

(Rana et al., 2025) reported that the IMLE Policy has competitive or favorable accuracy and sample efficiency in comparision to diffusion- and flow-based policies in visuomotor settings, while enjoying markedly lower test time due to the lack of iterative denoising or Ordinary Differential Equation

(ODE) integration. However, extending IMLE to *conditional* policies, where the output distribution depends on a sensory input context, poses two key challenges. First, prior methods (Rana et al., 2025) typically apply rejection sampling *per sample*, comparing generated trajectories only to the paired ground truth. This violates the **batch-global coverage** principle central to IMLE, often causing the policy to regress toward *averaged or midpoint behaviors* that fail to represent distinct modes in multimodal settings.

Second, closed-loop robotic tasks require strong **temporal smoothness** (ordered *sequences* of actions executed over time) across action sequences. Local smoothing heuristics (e.g., selecting candidates closest to the recent action history) have been used to mitigate mode-switching[1] during execution (Rana et al., 2025), but such methods remain brittle. They introduce mode commitment early in execution, lack global trajectory reasoning, and are sensitive to initial suboptimal choices or re-planning resets.

We introduce **Performer RS–IMLE for Multisensory Imitation** (PRISM), a **real-time multisensory imitation policy** that addresses both issues via a batch-global RS-IMLE objective and a temporal attention backbone. The model observes a short history of sensory tokens (RGB, depth, tactile, proprioception; when available) and uses a bank of *learned action query tokens*[2] (one per future timestep) to attend to this history through **bidirectional** linear attention (FAVOR$^+$ (Choromanski et al., 2020); see §4.2). In a single batched forward pass, these queries produce $K$ latent-conditioned action sequences. During training, an EMA-calibrated, batch-global rejection mask ensures that each candidate is sufficiently distinct from all ground truth sequences in the minibatch, preserving diversity without relying on iterative sampling. **The key to smooth trajectory generation lies in our receding horizon inference strategy**: at each timestep, the model generates $K$ action sequences, selects the most coherent, executes only the first $T_a$ actions, then slides the observation window forward and replans. This continuous replanning mechanism, combined with the single-pass generation efficiency, produces trajectories with $50\times$ reduced jerk compared to iterative sampling methods while maintaining real-time performance. Unlike diffusion or flow-based models, no iterative refinement or numerical integration is required.

**Contributions.**

- We present PRISM (Performer RS–IMLE for Multisensory Imitation), a **single-pass policy** that generates full action sequences using temporal cross-attention with linear-time Performer attention and bidirectional self-attention over learned query tokens.

- We introduce a **batch-global RS-IMLE** training objective objective with EMA-calibrated $\varepsilon$ and robust sequence distances, encouraging coverage across minibatch targets without adding test-time sampling steps; an optional small top-$K'$ coverage term further improves diversity.

- A time-preserving fusion encoder integrates RGB (wrist and, when present, static view), depth, tactile, proprioception, and audio. Modality-dropout studies identify gripper RGB and proprioception as essential signals, while some visual channels (e.g., depth in our setup) can be redundant, yielding graceful degradation under missing sensors.

- We report real-robot results on a Unitree GO2 with a D1 7-DoF arm and parallel gripper, using wrist and shoulder RGB, proprioception, and (when enabled) tactile and audio, across three loco-manipulation tasks: pre-manipulation parking, peg insertion, and pick-and-place.

- Across CALVIN, Meta-World, Robomimic, and the hardware suite, PRISM matches or exceeds diffusion-, flow-, and prior IMLE-based policies at matched wall-time. On CALVIN with 10% data, PRISM improves Success rate by $\sim$10% with reduced jerk by about $20\times$ over IMLE, $\sim$20% over flow matching, and $\sim$28% over diffusion. On Meta-World, it achieves 4-5% gains on Easy/Medium tasks and 5-12%improvement over state-of-the-art flow and diffusion policies on harder tasks. Real-world tests show 10-25% higher success rates across loco-manipulation scenarios, scaling with demonstration count.

---

[1]Here, a *mode* corresponds to a distinct high-probability strategy in the expert distribution (e.g., different grasps or paths), and undesirable *mode switching* occurs when the policy abruptly jumps between such valid strategies over time (Rhinehart et al., 2019; Mercat et al., 2020; Pham et al., 2023).

[2]*Learned action queries* are trainable vectors internal to the policy, each query corresponds to one future timestep and is refined by self- and cross-attention.

The remainder of this paper is organized as follows: Section 2 focuses on related works. Section 3 positions PRISM within the landscape of generative imitation. Section 4 formalizes our methodology, deriving the batch-global RS-IMLE objective from first principles and establishing its theoretical consistency. Section 5 presents empirical results across simulation and hardware, analyzing performance, computational efficiency, and failure modes. Theoretical proofs and extensive ablations are detailed in the Appendices.

## 2 RELATED WORKS

Behavior cloning (BC) maps observations to actions via supervised learning (Zhang et al., 2018; Florence et al., 2019), enabling fast single-pass inference for real-time control. Yet, naïve BC with mean squared error often fails in multimodal or contact-rich tasks, averaging diverse behaviors into low-precision outputs. Discretization reformulates control as classification (Zeng et al., 2022; Wu et al., 2020), but scales poorly in high dimensions. Mixture density networks and behavior transformers (Mandlekar et al., 2021b; Shafiullah et al., 2022) better capture multimodality but suffer from collapse and instability (Florence et al., 2021). Energy-based implicit models improve precision in multi-valued mappings (Florence et al., 2021), yet explicit BC struggles with mode coverage. PRISM addresses this via batch-global RS-IMLE, promoting diverse modes without averaging, while preserving single-pass efficiency.

Deep generative models, particularly diffusion and flow-based methods, have advanced visuomotor policy learning by modeling complex multimodal actions (Chi et al., 2023; Song et al., 2022; Liu et al., 2022). Diffusion models achieve stability and multimodality through iterative denoising, with Diffusion Policy setting state-of-the-art manipulation results (Chi et al., 2023), but their multi-step inference introduces latency unsuited for real-time control. Accelerations such as DDIM (Song et al., 2022), rectified flows (Liu et al., 2022), SE(3)-equivariant transformers (Funk et al., 2024), and adaptive solvers (Hu et al., 2024) improve efficiency, while Falcon diffusion (Chen et al., 2025) and Consistency Policy (Prasad et al., 2024) distill or reuse steps to reduce cost. Yet, ensuring smoothness and mode coverage under fewer steps remains difficult. PRISM instead achieves single-pass inference with batch-global rejection sampling, preserving multimodality and temporal coherence without iterative denoising.

Single-step generative policies avoid the cost of iterative sampling. GAN-based methods (Goodfellow et al., 2014; Chen et al., 2023) enable efficient inference but often collapse modes, limiting multimodal coverage in visuomotor control. IMLE instead matches each demonstration to a nearest sample, ensuring distributional coverage without adversarial training (Li & Malik, 2018), while RS-IMLE improves alignment via rejection sampling (Vashist et al., 2024). Yet, existing IMLE variants rely on local heuristics for temporal coherence, leading to mode switching and instability in long-horizon tasks. PRISM addresses this with a batch-global RS-IMLE objective and temporal cross-attention over multisensory inputs, producing multiple coherent trajectory candidates and avoiding premature mode commitment or costly iterative rejection.

Multimodal inputs such as RGB, depth, proprioception, tactile, and audio improve policy robustness by mitigating partial observability and noise (Avigal et al., 2022; Yu et al., 2025). Prior works fuse modalities through concatenation or attention, though few analyze their relative importance under sensor dropout. Temporal smoothness is also essential to prevent jitter from mode switching (Rhinehart et al., 2019; Mercat et al., 2020). Attention-based policies enhance long-horizon coherence, with linear-time variants like Performer enabling efficient $\mathcal{O}(T, m)$ reasoning (Choromanski et al., 2020). PRISM fuses per-timestep modalities into fixed-width tokens and applies bidirectional linear attention for real-time, non-autoregressive generation, with ablations showing gripper RGB and proprioception as critical while depth can be redundant.

## 3 PROBLEM STATEMENT

**Setting.** We consider imitation learning from demonstrations with temporally aligned multisensory observations. Each episode provides synchronized sensor streams comprising one or more RGB cameras (always wrist; optionally static view), and optionally depth, tactile, proprioception, and audio. Let $\mathcal{D} = \{(\mathbf{O}^{(n)}, \mathbf{A}^{(n)})\}_{n=1}^{N}$ denote $N$ episodes, where $\mathbf{O}^{(n)}$ and $\mathbf{A}^{(n)}$ are sequences of observations and actions.

**Observations and actions.** At discrete time $t$, the observation $\mathbf{o}_t$ concatenates all available sensors. The action is $\mathbf{a}_t \in \mathbb{R}^{D_a}$ with each dimension normalized to $[-1, 1]$, where $D_a = 7$ for tabletop

manipulation (end-effector deltas + gripper) and $D_a = 14$ for loco-manipulation (absolute pose + base velocities + gripper). Full parameterization details are in Appendix A.1.7.

**Windows.** For observation horizon $T_o$ and prediction horizon $T_p$:

$$\mathbf{O}_{t-T_o+1:t} = \{\mathbf{o}_{t-T_o+1}, \ldots, \mathbf{o}_t\}, \quad \mathbf{A}_{t+1:t+T_p} = \{\mathbf{a}_{t+1}, \ldots, \mathbf{a}_{t+T_p}\}.$$

Our goal is to learn a single-forward-pass, multimodal policy:

$$\hat{\mathbf{A}}_{t+1:t+T_p} \sim \pi_\theta(\cdot \mid \mathbf{O}_{t-T_o+1:t}, z), \tag{1}$$

where $z \sim \mathcal{N}(\mathbf{0}, \mathbf{I})$ is a latent variable.

**Assumptions.** (i) Sensors are time-aligned; missing modalities may occur. (ii) Actions are bounded and normalized. (iii) Demonstrations are multimodal: multiple valid action sequences may correspond to the same context.

## 4 PROPOSED APPROACH

Our goal is to learn a policy that is **multimodal** (capturing diverse behaviors), **temporally consistent** (generating smooth trajectories), and **efficient** (running in real-time). To achieve this, we introduce PRISM, a single-pass framework that decouples the problem into two stages: a) **Temporal Multi-Sensory Encoder** that fuses heterogeneous inputs while preserving the temporal dimension $(B, T, \text{features})$, b) **Single-Pass Generator** that produces full action sequences in parallel using bidirectional linear attention (FAVOR$^+$ (Choromanski et al., 2020)), and c) **Batch-Global RS-IMLE** training objective that provides a theoretically grounded mechanism for mode coverage without the inference cost of iterative diffusion. Figure 1 provides an overview. Pseudocode for training and receding-horizon inference appears in Alg. 1 and Alg. 2; stabilized FAVOR$^+$ features are detailed in Algorithm 3 (Appendix A.2).

### 4.1 TEMPORAL MULTI-SENSORY ENCODER

Let $T_o$ be the context horizon and $d$ the model width. For each timestep $t \in \{1, \ldots, T_o\}$ we form per-modality embeddings (wrist RGB, static RGB, depth, tactile, proprioception, audio, text; dataset-dependent) denoted $\{\mathbf{v}_t^{(m)}\}$ with dimensions $d_m$. We fuse *per timestep* to preserve temporal structure:

$$\mathbf{c}_t = \text{MLP}_{(\sum_m d_m) \to 1024 \to d}([\text{ concat of available } \mathbf{v}_t^{(m)} ]) \in \mathbb{R}^d, \tag{2}$$

and stack $\{\mathbf{c}_t\}$ to obtain context tokens $\mathbf{C} \in \mathbb{R}^{T_o \times d}$ with learned absolute positional embeddings. Per-timestep fusion avoids entangling temporal statistics, delegating long-range consistency to the generator. Modality usage per benchmark and detailed ablations appear in Appendix A.1.6 (Table 4) and B.1 (Figures 4–6); Section 5 validates that wrist RGB and proprioception are critical while depth is occasionally redundant.

### 4.2 SINGLE-PASS LINEAR-ATTENTION GENERATOR (BIDIRECTIONAL)

Given $\mathbf{C}$, the generator produces a length-$T_p$ action sequence in one forward pass. We initialize $T_p$ learnable query tokens $\mathbf{Q} \in \mathbb{R}^{T_p \times d}$, add a projected latent $z \sim \mathcal{N}(\mathbf{0}, \mathbf{I})$ (the stochastic seed enabling diverse trajectory candidates) and positional encodings $\mathbf{PE}_{pos}$, then apply $L$ Transformer blocks with (i) **bidirectional self-attention** over $\mathbf{Q}$ (no causal mask) and (ii) **cross-attention** to the full context $\mathbf{C}$ (also without causal masking). Both attentions use FAVOR$^+$ (linearized softmax) (Choromanski et al., 2020) with stabilized positive features, reducing cost from $\mathcal{O}(T^2)$ to $\mathcal{O}(T\,m)$ per head (full details in Algorithm 3, Appendix A.2):

$$\text{Attn}(\mathbf{Q}, \mathbf{K}, \mathbf{V}) \approx \frac{\Phi(\mathbf{Q})(\Phi(\mathbf{K})^\top \mathbf{V})}{\Phi(\mathbf{Q})(\Phi(\mathbf{K})^\top \mathbf{1}) + \varepsilon_a}, \qquad \Phi(\mathbf{X}) = \frac{\exp(\mathbf{X}W - \text{rowmax}(\mathbf{X}W))}{\sqrt{m}}, \tag{3}$$

where $W \sim \mathcal{N}(0, 1/\sqrt{d})$, $m$ random features, and $\varepsilon_a > 0$ clamps the denominator. A linear head maps final tokens to actions $\hat{\mathbf{A}} \in \mathbb{R}^{T_p \times D_a}$, where $D_a \in \{7, 14\}$ matches Sec. 3.

**Why bidirectional and linear?**   The policy is *non-autoregressive*, choosing the entire sequence jointly via bidirectional self-attention over $\mathbf{Q}$ and full cross-attention to $\mathbf{C}$. FAVOR$^+$ enables real-time control: Theorem 1 in Choromanski et al. (2020) proves unbiased softmax estimates with variance $O(1/m)$, preventing error accumulation over long horizons (formal analysis in Appendix E.4). Ablations (Figure 8b, Appendix B.2) show higher random-feature budgets improve accuracy; FAVOR$^+$ achieves 1.5–2$\times$ speedup vs. standard attention at equal wall-time (Table 7, Appendix B.5).

### 4.3 ROBUST SEQUENCE DISTANCE

We compare predicted and target sequences with a Charbonnier penalty $\varepsilon_c$ and per-dimension weights $w_d$ (inverse empirical scale), normalized over time to keep values $O(1)$:

$$D_\rho(\hat{\mathbf{A}}, \mathbf{A}) \;=\; \frac{1}{T_p} \sum_{t=1}^{T_p} \sum_{d=1}^{\mathbf{D_a}} w_d \sqrt{(\hat{a}_{t,d} - a_{t,d})^2 + \varepsilon_c^2}, \tag{4}$$

with $\varepsilon_c = 10^{-6}$. This metric is used both for training and for candidate selection at evaluation (when a learned proxy is not available), and its $1/T_p$ normalization keeps the rejection sampling threshold scale consistent across horizons. The Charbonnier formulation provides robustness to outliers while maintaining differentiability, critical for stable gradient flow in the IMLE objective (Barron, 2019).

### 4.4 BATCH-GLOBAL RS–IMLE OBJECTIVE

For each item $i$ in batch $B$, we draw $K$ latents yielding candidates $\{\hat{\mathbf{A}}_i^{(k)}\}_{k=1}^K$. Let $D_{i,k} = D_\rho(\hat{\mathbf{A}}_i^{(k)}, \mathbf{A}_i)$ and define batch-global distances $D_{i,k \to j} = D_\rho(\hat{\mathbf{A}}_i^{(k)}, \mathbf{A}_j)$ to all targets $j$. We reject candidates too close to *any* target:

$$\mathbb{I}_{i,k}^{\text{rej}} \;=\; \mathbb{I}\Big[ \min_j D_{i,k \to j} < \varepsilon_{\text{RS}} \Big], \tag{5}$$

and minimize the hard IMLE loss with fallback:

$$\mathcal{L}_{\text{hard}} \;=\; \frac{1}{B} \sum_{i=1}^B \min_{k \in \mathcal{K}_i} D_{i,k}, \quad \mathcal{K}_i = \begin{cases} \{k : \mathbb{I}_{i,k}^{\text{rej}} = 0\}, & \text{if nonempty,} \\ \{1, \dots, K\}, & \text{otherwise.} \end{cases} \tag{6}$$

We calibrate $\varepsilon_{\text{RS}}$ via EMA of batch quantile $q \in [0.2, 0.35]$ with hard clamps $(\varepsilon_{\min}, \varepsilon_{\max}) = (10^{-4}, 0.2)$:

$$\tilde{\varepsilon} = \text{Quantile}_q\big(\{D_{i,k \to j}\}\big), \quad \varepsilon_{\text{RS}} \leftarrow \text{clip}\big(\alpha\,\varepsilon_{\text{RS}} + (1-\alpha)\,\tilde{\varepsilon},\ \varepsilon_{\min}, \varepsilon_{\max}\big). \tag{7}$$

Lemma E.2 (Appendix E.2) proves this estimator has variance $O(1/N)$ by the Glivenko-Cantelli theorem, with empirical validation in Figure 15.

A small top-$K'$ soft-coverage term encourages diversity:

$$\mathcal{L}_{\text{soft}} = -\frac{1}{B} \sum_{i=1}^B \log \sum_{k \in \text{Top}K'(D_{i,\cdot})} \exp(-D_{i,k}/\tau), \tag{8}$$

yielding $\mathcal{L} = \mathcal{L}_{\text{hard}} + \lambda_{\text{soft}}\mathcal{L}_{\text{soft}}$ with $\lambda_{\text{soft}} \ll 1$. We use $K' \ll K$ (typically $K' = 3$ for $K = 16$) to focus gradients on relevant modes. Lemma E.3 (Appendix E.3) establishes this is equivalent to entropy-regularized $k$-center, preventing mode collapse. The batch-global rejection prevents candidates from covering multiple nearby targets, preserving gradient signal for alternative modes without test-time cost. Theoretical foundations (IMLE's equivalence to kernel-smoothed likelihood maximization, detailed justifications for soft coverage and batch-global rejection) are provided in Appendix E.

**Why Soft-coverage term?**   The hard-min objective used in standard IMLE propagates gradients only through the single closest candidate. When $K$ is finite and the dataset contains multiple modes, this can bias optimization toward frequent or dense modes because they are more likely to produce the minimal distance. To mitigate this optimization bias, we introduce a mild soft log-sum-exp over

the top-$K'$ candidates. This does not alter the underlying IMLE formulation but provides a controlled relaxation in which several nearby candidates contribute gradient signal without averaging across modes. Empirically, in Figure 7a (Appendix B.2), decreasing this term consistently reduces performance in multi-modal settings. We therefore view soft coverage as a practical stabilization mechanism rather than a theoretical requirement.

**Why batch-global rejection?** Masking by equation 5 prevents a single candidate from trivially "covering" many nearby targets across the whole batch, which would otherwise bias gradients toward a unimodal midpoint. This preserves gradient signal for alternative modes without increasing test-time cost. Sensitivity to $K$ and $\varepsilon_{\mathrm{RS}}$ appears in ablations (Figure 7, Appendix B.2), showing that performance saturates at $K \approx 16$ candidates. Because $D_{i,k \to j}$ is computed against all targets in the batch, RS–IMLE remains stable when multimodal clusters do not co-occur. In such cases the quantile naturally contracts to the batch's local density, avoiding suppression of candidates from absent modes.

## 4.5 RECEDING-HORIZON INFERENCE (SINGLE PASS)

At test time we observe $\mathbf{O}_{t-T_o+1:t}$, encode $\mathbf{C}$, draw $K$ latents, and produce $K$ action sequences in one pass. We select a single trajectory using one of two observation-only rules: **(1) ProxyScore** (when available). If proprioception or wrist pose is observable (e.g., CALVIN, hardware), we compute a simple smoothness proxy: each candidate's first action induces a predicted next end-effector pose, and we select the candidate whose induced pose is closest to the observed pose (or to the previously executed action). This reduces abrupt mode switches while requiring no action labels. **(2) Deterministic tie-break** (when proxy unavailable). If the dataset does not expose proprioception or EE pose (e.g., Robomimic), ProxyScore cannot be computed. We then select the candidate whose first action is closest (L2) to the last executed action. This encourages temporal consistency without needing ground-truth actions. Both rules operate using only observations, never target actions. We execute the first $T_a$ actions of the selected trajectory and slide the window by $T_a$. Because generation is non-autoregressive and linear-attention based, batched evaluation yields near-constant wall-time for moderate $K$ on modern accelerators. Full hyperparameters, preprocessing, and stability settings are provided in Appendices A.3 and A.1.6.

## 5 EXPERIMENTS

We evaluate PRISM on four established benchmarks: MetaWorld (50 tasks) (Yu et al., 2020), CALVIN (multi-sensory manipulation) (Mees et al., 2022), Robomimic (PH) (Mandlekar et al., 2021a), and a real-robot loco-manipulation suite, to answer: **RQ1** does sequence-level consistency reduce jerk and mode switches beyond best-of-$K$? **RQ2** can a single-pass, linear-attention policy meet 30 Hz as horizons grow? **RQ3** does multi-sensory fusion (RGB, depth, tactile, proprioception) sustain performance under occlusions/dropouts? **RQ4** how does PRISM compare to diffusion/flow at matched wall-time? **RQ5** what is the effect of batch-global rejection, $\varepsilon$-calibration, and candidate count $K$? **RQ6** do sim trends carry to hardware? Full setup details, fairness controls, and comprehensive per-task breakdowns are provided in Appendices B–C.

**MetaWorld (RQ1, RQ2, RQ4).** Under identical one-step inference (NFE= 1), PRISM improves success by **4–5%** on the Easy/Medium splits and by **5–12%** on the Hard/Very-Hard splits over the strongest prior one-step flow baseline (Table 1). Table 8 (Appendix C.1) provides the complete 50-task breakdown, including comparisons against DP3 (Ze et al., 2024), ManiCM (Lu et al., 2024), SDM (Jia et al., 2024), and AdaFlow (Hu et al., 2024), showing PRISM achieves the highest success rate across all difficulty categories. Relative to Diffusion Policy with 10-step denoising, gains reach $\sim$**12%** on the hardest tasks. These margins align with lower jerk/fewer mode switches (RQ1): bidirectional sequence generation avoids myopic, stepwise corrections that iterative samplers often need in contact-rich, multi-phase goals. Because PRISM is single-pass with linear attention, it holds 30 Hz when $T_o/T_p$ increase (RQ2), whereas multi-step policies must trade accuracy for latency (latency analysis in Table 7, Appendix B.5).

**CALVIN (RQ1–RQ3–RQ4)** Training on just **10%** of Env D, PRISM lifts success by $\sim$**10%** over IMLE Policy, $\sim$**28%** over Diffusion Policy, and $\sim$**20%** over Flow Matching Policy at matched budgets (Table 2). Motion quality improves markedly: jerk is reduced by $\sim$**20**$\times$ (from 1.05 to

---

**Algorithm 1** PRISM training: batch-global RS-IMLE with single-pass linear-attention generator

---

**Require:** Dataset $\mathcal{D} = \{(\mathbf{O}^{(n)}, \mathbf{A}^{(n)})\}$; horizons $(T_o, T_p)$; candidates $K$; model width $d$; heads $h$; FAVOR$^+$ features $m$

**Require:** Action weights $\{w_d\}_{d=1}^{D_a}$; Charbonnier $\varepsilon_c$; linear-attention clamp $\varepsilon_a$

**Require:** RS quantile $q \in [0.2, 0.35]$, EMA momentum $\alpha \in [0, 1)$, clamps $(\varepsilon_{\min}, \varepsilon_{\max}) = (10^{-4}, 0.2)$

**Require:** Optional soft coverage $(K', \tau, \lambda_{\text{soft}} \ll 1)$

1: Initialize encoder $E_\phi$, generator $G_\theta$ (with fixed $h, m$), optimizer; initialize $\varepsilon_{\text{RS}} > 0$

2: **for each** minibatch $\{(\mathbf{O}_i, \mathbf{A}_i)\}_{i=1}^B \sim \mathcal{D}$ **do**

3:     **Form windows:** for each $i$, take context $\mathbf{O}_{i,t-T_o+1:t}$ and target $\mathbf{A}_{i,t+1:t+T_p}$

4:     **Preprocess:** RGB/tactile $\to [0, 1]$; depth/proprio/audio $\to$ `float32`

5:     **Encode:** $\mathbf{C}_i \leftarrow E_\phi(\mathbf{O}_{i,t-T_o+1:t}) \in \mathbb{R}^{T_o \times d}$

6:     **Sample latents:** draw $z_{i,k} \sim \mathcal{N}(\mathbf{0}, \mathbf{I})$ for $k = 1..K$

7:     **Single batched forward:** $\hat{\mathbf{A}}_i^{(k)} \leftarrow G_\theta(\mathbf{C}_i, z_{i,k}) \in \mathbb{R}^{T_p \times D_a}$     ▷ bidirectional self & cross attention; FAVOR$^+$

8:     **Per-item robust distances:** $D_{i,k} \leftarrow \frac{1}{T_p} \sum_{t=1}^{T_p} \sum_{d=1}^{D_a} w_d \sqrt{(\hat{a}_{i,t,d}^{(k)} - a_{i,t,d})^2 + \varepsilon_c^2}$

9:     **Batch-global distances:** compute $D_{i,k \to j} \leftarrow D_\rho(\hat{\mathbf{A}}_i^{(k)}, \mathbf{A}_j)$ for all $i, k, j$

10:    **Quantile calibration:** $\tilde{\varepsilon} \leftarrow \text{Quantile}_q(\{D_{i,k \to j}\})$; $\varepsilon_{\text{RS}} \leftarrow \text{clip}(\alpha\,\varepsilon_{\text{RS}} + (1-\alpha)\,\tilde{\varepsilon}, \varepsilon_{\min}, \varepsilon_{\max})$   ▷ Lemma E.2

11:    **RS mask:** $\mathbb{I}_{i,k}^{\text{rej}} \leftarrow \mathbb{I}\big[\min_j D_{i,k \to j} < \varepsilon_{\text{RS}}\big]$

12:    **Hard IMLE loss:** $\mathcal{K}_i \leftarrow \{k : \mathbb{I}_{i,k}^{\text{rej}} = 0\}$; if $\mathcal{K}_i = \emptyset$ set $\mathcal{K}_i = \{1..K\}$; $\mathcal{L}_{\text{hard}} \leftarrow \frac{1}{B} \sum_{i=1}^B \min_{k \in \mathcal{K}_i} D_{i,k}$ ▷ Lemma E.1

13:    **if** *soft coverage* enabled **then**

14:       **Soft term:** $\mathcal{L}_{\text{soft}} \leftarrow -\frac{1}{B} \sum_i \log \sum_{k \in \text{Top}K'(D_{i,\cdot})} \exp(-D_{i,k}/\tau)$; $\mathcal{L} \leftarrow \mathcal{L}_{\text{hard}} + \lambda_{\text{soft}} \mathcal{L}_{\text{soft}}$   ▷ Lemma E.3

15:    **else**

16:       $\mathcal{L} \leftarrow \mathcal{L}_{\text{hard}}$

17:    **end if**

18:    **Update:** $\theta, \phi \leftarrow \theta, \phi - \eta\,\nabla_{\theta,\phi}\mathcal{L}$     ▷ grad clip; AMP-safe

19:    **Log:** rejection rate $\frac{1}{BK} \sum_{i,k} \mathbb{I}_{i,k}^{\text{rej}}$, current $\varepsilon_{\text{RS}}$, soft/hard loss ratio

20: **end for**

---

**Algorithm 2** Receding-horizon inference with single-pass candidate selection

---

**Require:** Trained $E_\phi, G_\theta$; horizons $(T_o, T_p, T_a)$; candidates $K$; current time index $t$

1: **Observe & preprocess:** $\mathbf{O}_{t-T_o+1:t}$; RGB/tactile$\to [0, 1]$, depth/proprio/audio$\to$`float32`

2: **Encode context:** $\mathbf{C} \leftarrow E_\phi(\mathbf{O}_{t-T_o+1:t}) \in \mathbb{R}^{T_o \times d}$

3: **Single batched generation:** sample $z_k \sim \mathcal{N}(\mathbf{0}, \mathbf{I})$ and compute $\hat{\mathbf{A}}^{(k)} \leftarrow G_\theta(\mathbf{C}, z_k)$ for $k = 1..K$   ▷ bidirectional self & cross attention; FAVOR$^+$

4: **Select trajectory:** $k^\star \leftarrow \arg\min_k \text{ProxyScore}(\hat{\mathbf{A}}^{(k)}, \mathbf{O}_{t-T_o+1:t})$ if proxy available; else use deterministic tie-break; see Sec. 4.5

5: **Execute & slide:** apply $\hat{\mathbf{A}}_{1:T_a}^{(k^\star)}$; set $t \leftarrow t + T_a$; append new observations; repeat

---

0.05) and mode switches by $\sim$**3**$\times$ (from 0.28 to 0.10), indicating that batch-global RS–IMLE preserves diverse alternatives while maintaining sequence coherence (RQ1/RQ4). Under occlusion and single-modality dropouts, PRISM loses only **3–5%** versus nominal performance, whereas baselines degrade by 8–15% (Table 2). Detailed single-modality dropout analysis (Figure 4, Appendix B.1) shows removing wrist RGB or proprioception causes the most severe degradation (41.5% and 15.8% success respectively), while depth dropout has minimal impact (66.2% success, nearly identical to the no-dropout baseline of 65.2%). Pairwise modality ablations (Figures 5–6, Appendix B.1) confirm that the combination of wrist RGB and proprioception is critical, with their simultaneous removal leading to near-complete failure. These results validate RQ3: per-timestep fusion consistently extracts the most from wrist RGB and proprioception, while depth is occasionally redundant in our setup. Time-matched multi-step samplers (Diffusion Policy with NFE=10, Flow Matching Policy with NFE=5) remain slower and jerkier even when wall-clock time is controlled (detailed comparison in Table 7, Appendix B.5).

**Robomimic (PH) (RQ4).** On image-based tasks, PRISM remains *single-pass* yet matches or surpasses several multi-step policies on average success (Table 3). Against strong one-step baselines

Table 1: Performance benchmarks with 3D input. MetaWorld (50 tasks). One-step (NFE= 1) unless noted.One-step (NFE= 1) unless noted. Input: wrist RGB + proprioception; depth rendered from MuJoCo simulator Higher Success rate (%) is better. See Table 8 (Appendix C.1) for complete per-task breakdown.

| Method | NFE ↓ | Easy (28) ↑ | Medium (11) ↑ | Hard (6) ↑ | Very Hard (5) ↑ |
|---|---|---|---|---|---|
| Diffusion Policy (Chi et al., 2023) | 10 | 83.6 | 31.1 | 9.0 | 26.6 |
| Flow Matching Policy (Black et al., 2024) | 1 | 61.4 | 20.6 | 13.4 | 36.0 |
| IMLE Policy (Rana et al., 2025) | 1 | 75.6 | 60.4 | 43.5 | 76.0 |
| PRISM | **1** | **96.4** | **85.5** | **58.0** | **85.8** |

Table 2: CALVIN — success, robustness, and motion quality (mean $\pm$ s.e.; $K$=8). Input: wrist RGB + static RGB + depth + tactile + proprioception (10% of Env D). Modality dropout details in Figures 4–6 (Appendix B.1).

| Method | No Dropout ↑ | RGB Dropout ↑ | Depth Dropout ↑ | Tactile Dropout ↑ | Jerk ↓ | Mode Switch Rate ↓ |
|---|---|---|---|---|---|---|
| Diffusion Policy(Chi et al., 2023) | 36.4 $\pm$ 9.2 | 38.0 $\pm$ 3.5 | 34.4 $\pm$ 7.5 | 35.2 $\pm$ 7.2 | 3.7834 | 0.5150 |
| Flow Matching Policy(Black et al., 2024) | 44.4 $\pm$ 3.7 | 45.6 $\pm$ 2.7 | 45.6 $\pm$ 4.1 | 42.0 $\pm$ 3.5 | 1.1482 | 0.5857 |
| IMLE Policy(Rana et al., 2025) | 56.2 $\pm$ 6.5 | 55.2 $\pm$ 6.2 | 51.2 $\pm$ 10.3 | 54.2 $\pm$ 5.7 | 1.0527 | 0.2764 |
| PRISM | **65.2 $\pm$ 4.3** | **62.0 $\pm$ 5.8** | **66.2 $\pm$ 4.8** | **62.4 $\pm$ 5.7** | **0.0519** | **0.1006** |

(Consistency Policy (Prasad et al., 2024)) and prior IMLE variants, PRISM is ahead by **2–5%** on average; versus AdaFlow (NFE=$\sim$1.2), PRISM is competitive. Table 9 (Appendix C.1) provides the complete comparison including RectifiedFlow (Liu et al., 2022), ActionFlow (Funk et al., 2024), Falcon-DDPM (Chen et al., 2025), and other multi-step baselines. This supports that coherent, sequence-level generation can substitute for iterative refinement at comparable wall-time. We note that AdaFlow (NFE=$\sim$1.2) slightly outperforms PRISM on PH tasks (95.6% vs. 92.2% average success). We attribute this to the near-unimodal nature of PH demonstrations and the absence of depth, tactile, or multi-view RGB, which limits the benefits of PRISM's multisensory encoder. AdaFlow's variance-adaptive step-size is specifically optimized for unimodal accuracy, whereas PRISM excels when multimodality is present, as evidenced by the superior performance on CALVIN (Table 2) and MetaWorld Hard/Very-Hard splits (Table 1).

**Real-World Evaluation (RQ2–RQ6)** We deploy PRISM on a Unitree Go2 with a D1 arm and parallel gripper, using history $H$=8, horizon $T$=16, chunk $T_a$=8, and $K$=5 candidates/step. PRISM outperforms baselines by **10–25%** across pre-manipulation parking, peg-in-hole, and pick-and-place tasks, with high inference speed (**RQ2**, **RQ6**). Performance scales with demonstration count: +10% (parking), +20% (peg-in-hole), and +30% (pick-and-place) when increasing from 15 to 35 demos as shown in Figure 3. Additional tabletop manipulation experiments (cup stacking, can picking) demonstrate consistent 15–20% performance advantages over baselines (Figure 12, Appendix D), with PRISM maintaining 50 Hz inference across all scenarios. Hardware setup details, including sensor configurations and task specifications, are provided in Appendix D (Figures 10–11). The consistent 10–25% improvement across all real-world tasks demonstrates that PRISM's benefits translate from simulation to hardware, with the 50 Hz sustained performance addressing **RQ2** (single-pass efficiency) in real-world constraints. The demonstration scaling (10–30% improvement) shows that PRISM effectively leverages additional training data.

**Architecture (RQ2, RQ5).** Comprehensive ablations (Figures 8–7, Appendix B.2) show performance saturates at $H$=8 heads, benefits from increased random features up to $m$=512, and shows diminishing returns beyond $K$≈16 candidates. Batch-global rejection with EMA $\varepsilon$ calibration reduces midpoint collapse (lower mode-switch rate) and improves success; a small top-$K'$ coverage term ($\lambda_{\text{soft}}$≈0.02) adds a boost without hurting single-candidate success. As proven in Lemma E.2 (Appendix E.2), the batch-global quantile estimator variance decreases as $O(1/N)$. Figure 15 empirically validates this, showing PRISM maintains stable $\varepsilon_{\text{RS}}$ calibration across batch sizes 8–512, with coefficient of variation dropping below 5% at batch size 64. Note, we provide additional insights on the efficiency of PRISM and a failure analysis in Appendix. B.4.

## 6 LIMITATIONS

While PRISM offers a powerful framework for single-pass, multisensory visuomotor policy learning with significant improvements in accuracy, smoothness, and real-time control, several limitations

Table 3: Robomimic (PH) — success by task (mean ± s.e.); one-step unless noted (NFE shown). Input: wrist RGB + proprioception (image-based control, 200 demonstrations per task). Extended comparison with multi-step baselines in Table 9 (Appendix C.1).

| Method | NFE ↓ | Lift ↑ | Can ↑ | Square ↑ | Transport ↑ | Tool Hang ↑ | Average ↑ |
|---|---|---|---|---|---|---|---|
| Diffusion Policy(Chi et al., 2023) | 15 | 1.00 | 0.99 ± 0.01 | 0.92 ± 0.03 | 0.79 ± 0.04 | 0.55 ± 0.05 | 0.85 |
| Flow Matching Policy(Black et al., 2024) | 1 | 0.99± 0.01 | 0.96± 0.01 | 0.79± 0.02 | 0.66± 0.05 | 0.86± 0.05 | 0.85 |
| IMLE Policy (Rana et al., 2025) | 1 | 1.00 | 0.94 ± 0.01 | 0.81 ± 0.01 | 0.85 ± 0.05 | 0.75 ± 0.06 | 0.87 |
| PRISM | **1** | **1.00** | **1.00** | 0.84 ± 0.02 | **0.91 ± 0.02** | **0.86 ± 0.03** | **0.922** |

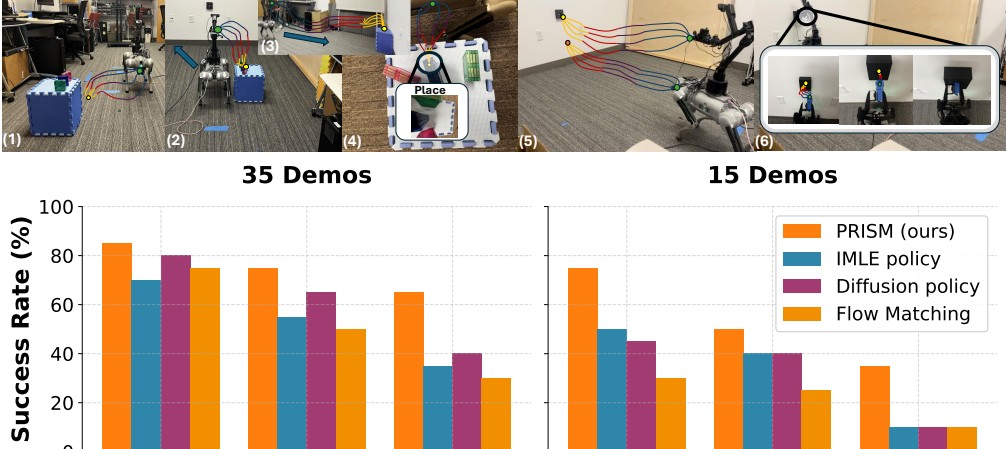

Figure 3: **Hardware setup and performance. Top:** Real-world platform and tasks. Unitree GO2+D1 arm with wrist/shoulder RGB (30 Hz), optional tactile and audio. We evaluate premanipulation parking (1-2), pick-and-place (3-4), and insertion (5-6). **Bottom:** Real-world locomanipulation success (50 trials/method). PRISM achieves higher success and mode coverage while maintaining 30 Hz. Additional manipulation tasks in Figure 12 (Appendix D).

remain. We do not use pretrained encoders to ensure fair comparison with diffusion, flow, and IMLE baselines. Incorporating pretrained visual or multi-modal encoders is a promising direction for future work, especially for language- or audio-conditioned tasks. PRISM relies on a batch–global $\varepsilon_{RS}$ calibrated from minibatch distances, which couples samples and can be sensitive to batch size and dataset heterogeneity; miscalibration risks either under-rejection (mode averaging) or over-rejection (slow optimization). Developing adaptive, data-driven rejection thresholds could mitigate this sensitivity in future work. Lastly, persistent sensor failures present challenges despite the encoder's robustness to intermittent dropout; integrating explicit uncertainty estimation by taking advantage of our fast inference time through ensemble encoders or multiple latent draws or online sensor diagnostics could further enhance resilience.

## 7 CONCLUSION

We presented PRISM, a single-pass visuomotor policy that fuses multisensory history, generates full action sequences with bidirectional Performer attention, and preserves multimodal coverage via batch–global RS–IMLE. Across benchmarks, PRISM is accurate, smooth, and fast: on MetaWorld (NFE= 1) it improves over strong flows by **4-5%** on average and by up to ∼**5%** on Hard/Very-Hard, and exceeds diffusion variants by ∼**20%**; on CALVIN, using only **10%** of the data, it outperforms IMLE-Policy by ∼**10%**, flow by ∼**20%**, and diffusion by ∼**25%** with markedly lower jerk; and on real loco-manipulation and manipulation, it improves success by **10-25%** while sustaining real-time inference. Future work includes adaptive rejection control, and learned feature maps for linear attention, and RL fine-tuning for generalization.

## 8 REPRODUCIBILITY

Anonymized code and demonstration datasets will be made available on our project webpage: https://anonymous2271106-a11y.github.io/PRISM/.

We adopt widely-used benchmarks—**CALVIN**, **Robomimic**, and **MetaWorld**—to ensure ease of comparison and reproducibility across future work. Full implementation details, including hyperparameters, environment specifications, and real-world hardware setup, are included in the Appendix.

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

# A  EXPERIMENTAL SETUP AND IMPLEMENTATION DETAILS

This appendix provides full details of the benchmarks, modalities, baseline implementations, and cross-benchmark differences used in all PRISM experiments. This section consolidates information that was previously scattered across Sections 4–5, and responds to reviewer requests for a unified, transparent description of how baselines were adapted for each dataset.

## A.1  BENCHMARK SPECIFICATIONS

We evaluate PRISM across four widely used visuomotor imitation-learning benchmarks: Meta-World, CALVIN, Robomimic (PH), and a real-world loco-manipulation suite. This section provides implementation details omitted from the main paper, including dataset preprocessing, modality configurations, and baseline reproduction protocols. Our goal is to make experimental comparisons reproducible and consistent across settings.

### A.1.1  METAWORLD (50 TASKS)

We adopt the official MetaWorld MT50 benchmark (Yu et al., 2020), following the standard data-collection pipeline used in diffusion-, flow-, and IMLE-based visuomotor policies. We use wrist RGB images and proprioception for all experiments; depth is rendered from MuJoCo but not used during training unless specified. All demonstrations are frame-aligned and downsampled to 20–30 Hz. Observation windows use $T_o = 4$ and prediction horizons use $T_p = 16$. For baseline reproduction, we follow the settings in Diffusion Policy, Flow Matching Policy, IMLE Policy, ManiCM, AdaFlow, and DP3. Each baseline is trained using the same dataset splits and action parameterization (7D end-effector deltas + gripper). For one-step baselines (Flow Matching, IMLE, Consistency-FM), we verify their reported NFE configurations. Multi-step diffusion baselines use 10-step DDPM or Diffusion Poliy unless explicitly stated. This ensures fair wall-clock comparisons.

### A.1.2  CALVIN (MULTIMODAL MANIPULATION)

CALVIN provides wrist-camera RGB, depth, static-camera RGB, tactile, and proprioception (Mees et al., 2022). PRISM along with all the baselines uses all available modalities unless ablations specify dropouts. Baselines (Diffusion Policy, Flow Matching, IMLE Policy) are re-trained using the same observation and control rate as PRISM. All sensor streams are temporally aligned at 30 Hz and processed using the same normalization scheme across baselines. PRISM uses the temporal multisensory encoder to fuse all available per-timestep modalities; we do not introduce any dataset-specific architectural changes. Depth is included by default unless removed in ablations. Because CALVIN exposes proprioception, PRISM uses the ProxyScore candidate-selection rule during receding-horizon inference, allowing smooth and consistent rollouts without requiring access to ground-truth actions. Baseline methods (Diffusion Policy, Flow Matching Policy, and IMLE Policy) are re-trained under matched observation spaces, control rates, and action parameterization to ensure fair comparison.

### A.1.3  ROBOMIMIC (PROFICIENT HUMAN DEMONSTRATIONS)

We use the Proficient Human (PH) datasets for Lift, Can, Square, Transport, and Toolhang (Mandlekar et al., 2021b). All policies operate on wrist RGB and proprioception. Demonstrations are temporally aligned at 20 Hz. We follow the same evaluation protocol used by SDM(Jia et al., 2024), Consistency Policy(Prasad et al., 2024), evaluating 50 initialization seeds per task and reporting mean success. Baselines include Diffusion Policy, Consistency Policy, Consistent-FM, IMLE Policy, SDM, and ActionFlow. When baseline code differs in observation normalization (e.g., per-sequence vs. global statistics), we adapt preprocessing to match the respective paper.

### A.1.4  REAL-WORLD LOCO-MANIPULATION AND TABLETOP MANIPULATION

We evaluate PRISM on a Unitree GO2 equipped with a D1 7-DoF arm, wrist RGB, shoulder RGB, vision-based tactile sensing, audio, and proprioception. Sensor streams are time-aligned using ROS timestamps and synchronized to 30 Hz. We use three tasks—pre-manipulation parking, peg insertion, and pick-and-place—each using 15-35 teleoperated demonstrations. For all locomanipulation

tasks we use vision and tactile. For task shown in Figure 11 in the bottom row, we use vision and tactile as well. For top row task in Figure 11 we use vision and audio. We also added new experimets on language conditioned tasks shown in Figure 14, where we've used vision and text. Baselines are re-implemented following the same inference-time assumptions: Diffusion Policy (10-step denoising), Flow Matching (1-step), and IMLE Policy (1-step).

### A.1.5 BASELINE METHODS AND ABBREVIATIONS

To keep result tables compact, we use consistent abbreviations across benchmarks: "MW" denotes MetaWorld, "CV" denotes CALVIN, and "RM" denotes Robomimic (PH). All competing methods are re-trained under matched observation modalities, control rate, action parameterization, and data budgets. For multi-step baselines, we report main results using their recommended number of function evaluations (NFE). For one-step baselines, evaluation faithfully follows the settings described in Consistency-FM(Yang et al., 2024), ensuring that PRISM is compared against the strongest available accelerations. We reimplemented baselines only when their setup differed from ours; otherwise, we use originally reported numbers under matching conditions.

### A.1.6 DATASET DETAILS

Table 4 summarizes the modality availability across benchmarks. All datasets include wrist-mounted RGB as the primary visual input.

Table 4: Modality availability per benchmark. ✓: available and used; —: not available; $\Delta$: task-dependent.

| Benchmark | Wrist RGB | Static RGB | Depth | Tactile | Proprio | Audio |
|---|---|---|---|---|---|---|
| MetaWorld | ✓ | — | — | — | ✓ | — |
| CALVIN | ✓ | ✓ | ✓ | ✓ | ✓ | — |
| Robomimic | ✓ | — | — | — | ✓ | — |
| Real Hardware | ✓ | ✓ | — | $\Delta$ | ✓ | $\Delta$ |

**Data Preprocessing.**

- **Visual inputs**: Center-cropped and resized (200×200 for static, 84×84 for wrist), normalized to [0,1]
- **Depth**: Cast to `float32`, no normalization
- **Proprioception**: Per-dimension z-normalization using training statistics
- **Tactile**: 6-channel force/torque readings normalized to [0,1]
- **Temporal alignment**: All modalities synchronized to camera timestamps (30 Hz)

### A.1.7 ACTION SPACE PARAMETERIZATION

**Tabletop manipulation** ($D_a = 7$)**:**

- Translation increments: $(\Delta x, \Delta y, \Delta z)$ in meters
- Rotation increments: $(\Delta r_x, \Delta r_y, \Delta r_z)$ in radians (small-angle Euler)
- Gripper: continuous open/close command $\in [-1, 1]$

All increments are expressed in the robot base frame.

**Loco-manipulation** ($D_a = 14$)**:**

- Gripper command: $\in [-1, 1]$
- End-effector position: $(x, y, z)$ in meters
- End-effector orientation: $(q_w, q_x, q_y, q_z)$ unit quaternion
- Base linear velocity: $(v_x, v_y, v_z)$ in m/s (body frame)

- Base angular velocity: $(\omega_x, \omega_y, \omega_z)$ in rad/s (body frame)

Sensor observations are synchronized to camera timestamps at 30 Hz. Frame conventions follow each dataset's standard (see dataset-specific details in Section A.1.6).

### A.1.8 EVALUATION PROTOCOLS

Across all benchmarks, we follow the standard evaluation protocols used in prior visuomotor policy work. For MetaWorld, we evaluate all 50 tasks stratified into Easy/Medium/Hard/Very-Hard splits, using 10 demonstrations per task to measure data efficiency and reporting success as task completion within episode limits. CALVIN is evaluated under the multi-task setting using 10% of the Environment D dataset, with Success@K computed by selecting the best of $K$ candidates using task-specific proxy functions. For Robomimic, we use the Proficient-Human (PH) split with 200 demonstrations per task and perform image-based control on Lift, Can, Square, Transport, and Tool-Hang. Real-world evaluation consists of 50 trials per method under randomized initial conditions with a 60-second timeout, covering pre-manipulation parking, peg insertion, and pick-and-place tasks with varying object geometries.

### A.2 MODEL ARCHITECTURE

**Temporal Multi-Sensory Encoder.** The encoder processes each modality independently before temporal fusion. We utilize a ResNet-18 backbone for RGB inputs to leverage pre-trained features, while other modalities use lightweight custom encoders:

- **RGB streams** (wrist $\mathbf{v}_t^{\text{rgb-w}}$, static $\mathbf{v}_t^{\text{rgb-s}}$): ResNet-18 backbone (pre-trained on ImageNet), removing the final FC layer and projecting features to $d{=}512$.
- **Depth streams** (wrist $\mathbf{v}_t^{\text{dep-w}}$, static $\mathbf{v}_t^{\text{dep-s}}$): Three convolutional layers ($1{\rightarrow}64{\rightarrow}128{\rightarrow}256$ channels) with ReLU activations, adaptive pooling to $2{\times}2$, and linear projection to $d{=}256$.
- **Tactile** ($\mathbf{v}_t^{\text{tac}}$): Six-channel input processed through identical CNN architecture (3 layers), output dimension 256.
- **Proprioception** ($\mathbf{v}_t^{\text{prop}}$): Two-layer MLP (input$\rightarrow$128$\rightarrow$64) processing joint states and velocities.

Per-timestep fusion preserves temporal structure:

$$\mathbf{c}_t = \text{MLP}_{(\sum_m d_m)\rightarrow 1024 \rightarrow d}\left(\left[\mathbf{v}_t^{\text{rgb-s}}; \mathbf{v}_t^{\text{rgb-w}}; \mathbf{v}_t^{\text{dep-s}}; \mathbf{v}_t^{\text{dep-w}}; \mathbf{v}_t^{\text{tac}}; \mathbf{v}_t^{\text{prop}}\right]\right) \tag{9}$$

**Single-Pass Generator.** The generator employs $L{=}6$ transformer blocks with bidirectional self-attention over $T_p$ learned query tokens and cross-attention to context $\mathbf{C}$. Both attention mechanisms use FAVOR$^+$ linearization:

---

**Algorithm 3** FAVOR$^+$ Linear Attention Implementation

---

**Require:** Query $\mathbf{Q}$, Key $\mathbf{K}$, Value $\mathbf{V} \in \mathbb{R}^{L \times d_h}$
**Require:** Random features $\mathbf{W} \in \mathbb{R}^{d_h \times m}$, $W_{ij} \sim \mathcal{N}(0, 1/\sqrt{d_h})$
1: **function** STABILIZEDFEATURES($\mathbf{X}$)
2:     $\mathbf{U} \leftarrow \mathbf{XW}$
3:     $\mathbf{U} \leftarrow \mathbf{U} - \text{rowmax}(\mathbf{U})$            ▷ Numerical stabilization
4:     **return** $\exp(\mathbf{U})/\sqrt{m}$
5: **end function**
6: **function** LINEARATTENTION($\mathbf{Q}, \mathbf{K}, \mathbf{V}$)
7:     $\phi_Q \leftarrow$ STABILIZEDFEATURES($\mathbf{Q}$)
8:     $\phi_K \leftarrow$ STABILIZEDFEATURES($\mathbf{K}$)
9:     $\mathbf{S} \leftarrow \phi_K^\top \mathbf{V}$, $\mathbf{n} \leftarrow \phi_K^\top \mathbf{1}$
10:     **return** $(\phi_Q \mathbf{S}) \oslash (\phi_Q \mathbf{n} + \varepsilon_a)$
11: **end function**

---

**Comment on $\varepsilon_{\text{RS}}$ and $K$.** Because $D_{i,k \to j}$ is computed against all targets in the batch, RS–IMLE remains stable when multimodal clusters do not co-occur. In such cases the quantile naturally contracts to the batch's local density, avoiding suppression of candidates from absent modes.

## A.3 TRAINING CONFIGURATION

**Batch-Global RS-IMLE.** The training objective combines hard IMLE loss with optional soft coverage:

$$\mathcal{L}_{\text{hard}} = \frac{1}{B} \sum_{i=1}^{B} \min_{k \in \mathcal{K}_i} D_\rho(\hat{\mathbf{A}}_i^{(k)}, \mathbf{A}_i) \tag{10}$$

$$\mathcal{L}_{\text{soft}} = -\frac{1}{B} \sum_{i=1}^{B} \log \sum_{k \in \text{Top}_{K'}(D_{i,\cdot})} \exp(-D_{i,k}/\tau) \tag{11}$$

$$\mathcal{L}_{\text{total}} = \mathcal{L}_{\text{hard}} + \lambda_{\text{soft}} \mathcal{L}_{\text{soft}} \tag{12}$$

where $\mathcal{K}_i$ contains unrejected candidates based on batch-global distances, and the rejection threshold $\varepsilon_{\text{RS}}$ is dynamically calibrated via EMA of batch quantiles.

**Optimization Details.**

- **Optimizer**: AdamW with learning rate $10^{-4}$ (MetaWorld, Robomimic) or $5 \times 10^{-5}$ (CALVIN)
- **Batch size**: 128 with mixed precision training
- **Gradient clipping**: Optional at 1.0 for stability
- **Learning rate schedule**: Cosine annealing with warmup
- **RS-IMLE calibration**: Quantile $q \in [0.2, 0.35]$, EMA momentum $\alpha = 0.9$
- **Robust distance**: Charbonnier with $\varepsilon_c = 10^{-6}$, per-dimension weights from inverse training statistics

## A.4 MATHEMATICAL NOTATION

Table 5 provides a comprehensive reference for notation used throughout the paper.

Table 5: Mathematical notation and definitions.

| Symbol | Description |
|---|---|
| $\mathcal{D} = \{(\mathbf{O}^{(n)}, \mathbf{A}^{(n)})\}_{n=1}^{N}$ | Dataset of $N$ demonstration episodes |
| $\mathbf{o}_t \in \mathbb{R}^{D_o}$ | Multisensory observation at time $t$ |
| $\mathbf{a}_t \in [-1, 1]^{D_a}$ | Normalized action (7D tabletop, 14D loco-manipulation) |
| $T_o, T_p, T_a$ | Observation, prediction, and action execution horizons |
| $\mathbf{C} \in \mathbb{R}^{T_o \times d}$ | Temporally-fused context tokens |
| $\mathbf{Q} \in \mathbb{R}^{T_p \times d}$ | Learned query tokens for action generation |
| $G_\theta$ | FAVOR$^+$ transformer generator with parameters $\theta$ |
| $D_\rho(\cdot, \cdot)$ | Robust Charbonnier sequence distance |
| $\varepsilon_{\text{RS}}$ | Batch-global rejection threshold (EMA-calibrated) |
| $K, K'$ | Number of candidates; top-$K'$ for soft coverage |
| $m$ | Random features for FAVOR$^+$ linearization |

# B EXTENDED EXPERIMENTAL RESULTS AND ABLATIONS

This appendix provides comprehensive ablation studies and extended experimental results supporting the main paper claims, including modality robustness analysis, architectural design choices, and computational efficiency measurements.

## B.1 MODALITY ROBUSTNESS UNDER DROPOUT

We conduct extensive ablation studies on CALVIN to assess the contribution of each sensory modality. Figure 4 presents single-modality dropout results, demonstrating that wrist RGB and proprioception are critical for task success, while depth sensors can occasionally be redundant in our setup.

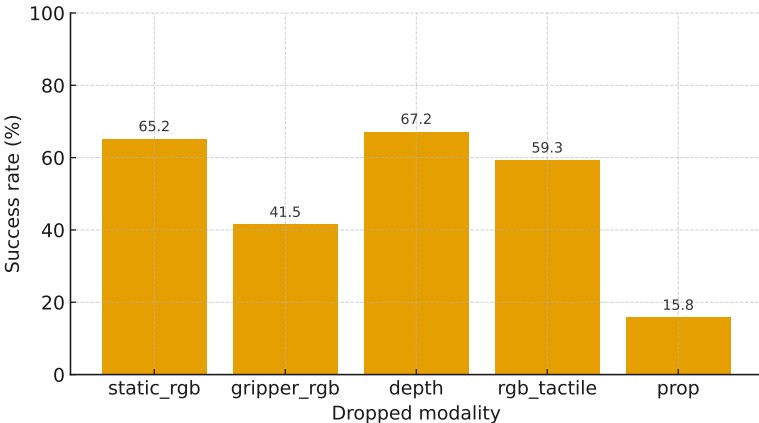

Figure 4: **Single-modality dropouts on CALVIN.** Success rate (%) when individual modalities are removed during evaluation. Wrist RGB and proprioception show the most significant impact, with performance dropping to 41.5% and 15.8% respectively when removed.

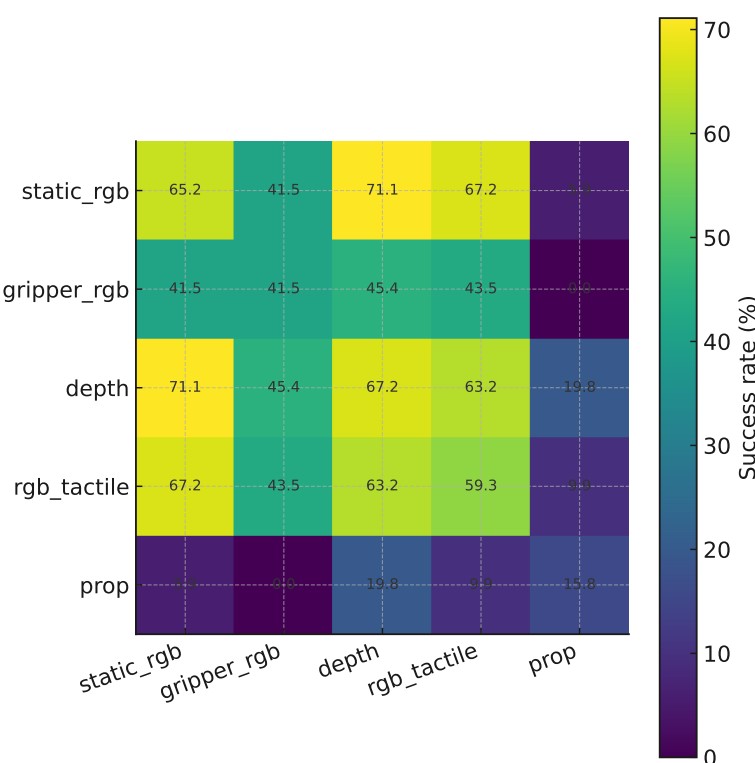

Figure 5: **Pairwise modality dropouts.** Success rate (%) when pairs of modalities are simultaneously removed. The combination of wrist RGB and proprioception removal leads to near-complete failure, confirming their critical role.

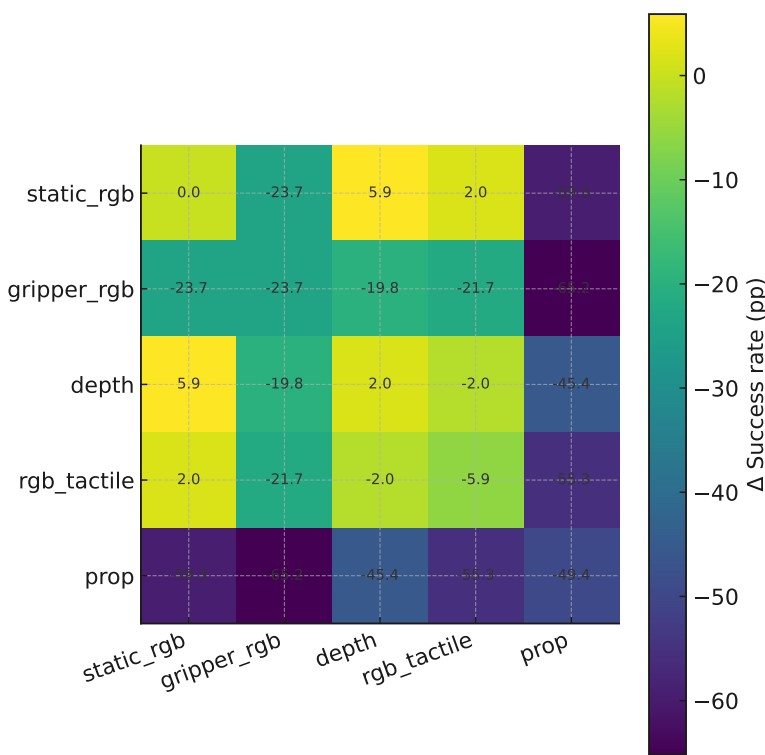

Figure 6: **Pairwise dropout impact relative to baseline.** Change in success rate (percentage points) compared to full modality performance. Negative values indicate performance degradation.

## B.2 ARCHITECTURE AND HYPERPARAMETER ABLATIONS

We conduct systematic ablations on CALVIN to validate our architectural choices and hyperparameter selections. Figure 8 presents comprehensive ablations of key architectural components (attention heads, FAVOR$^+$ features, transformer depth, candidate count $K$), while Figure 7 analyzes training hyperparameters (soft coverage weight, RS-IMLE calibration, observation horizon). These studies inform our default configuration choices and demonstrate the robustness of our approach across reasonable hyperparameter ranges.

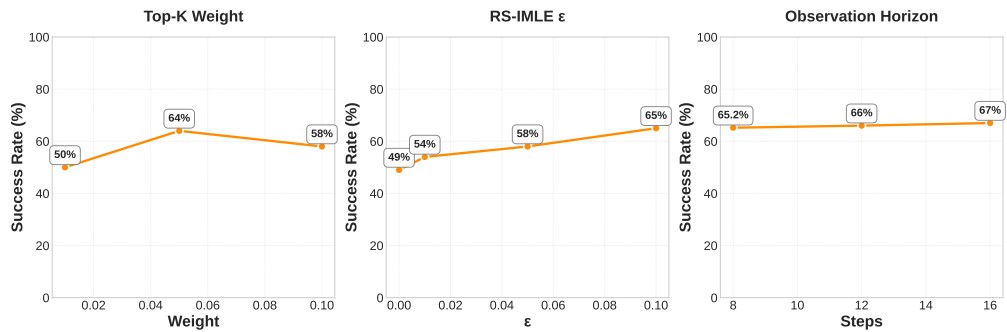

Figure 7: **Training hyperparameter ablations on CALVIN.** Effect of (a) soft coverage weight $\lambda_{\text{soft}}$, (b) batch-global RS-IMLE threshold calibration, and (c) observation horizon on performance. The EMA-calibrated threshold and small coverage term ($\lambda_{\text{soft}} = 0.02$) provide optimal trade-off between diversity and accuracy.

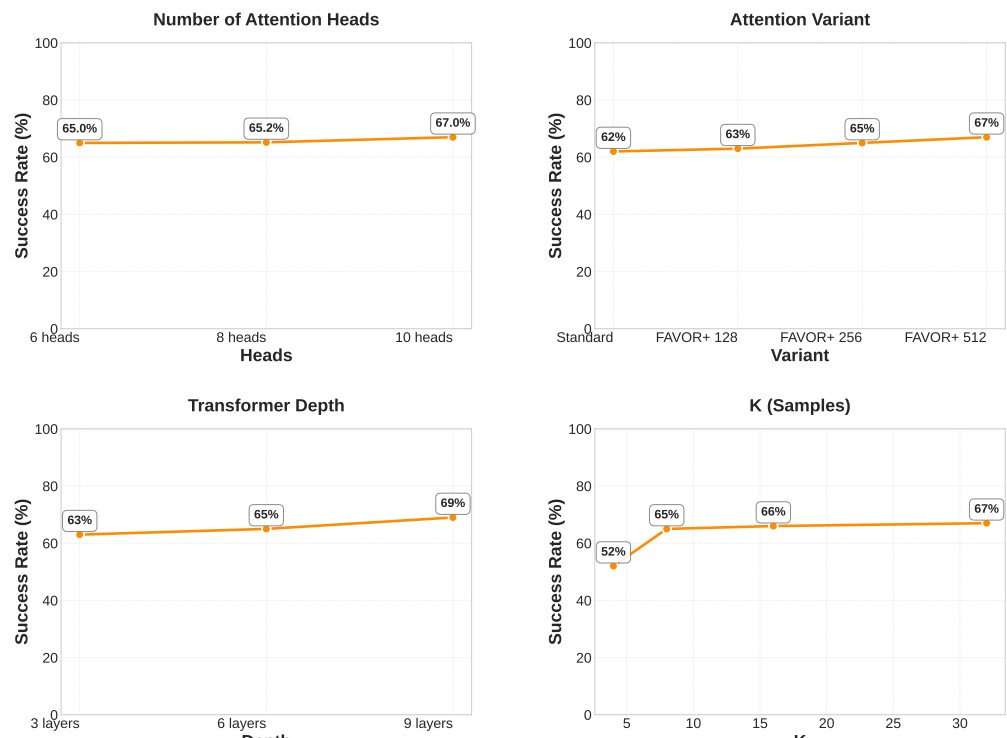

Figure 8: **Architecture ablations on CALVIN.** Impact of (a) attention heads, (b) FAVOR$^+$ random features, (c) transformer depth, and (d) number of latent candidates $K$ on success rate. Performance saturates at $H = 8$ heads, benefits from increased random features up to $m = 512$, improves with depth, and shows diminishing returns beyond $K = 16$ candidates.

### B.3 PUSH-T MULTIMODALITY STUDY

To visualize how different policies behave in states with varying inherent multi-modality, we follow the Push-T setup from Rana et al. (2025). We fix the T-shaped object and gradually sweep the robot end-effector start position along the top edge of the T. States near the corners are effectively unimodal (most demonstrations push in a consistent direction), whereas states near the center exhibit high multi-modality (demonstrations split between left and right pushes).

Figure 9 compares PRISM, the UNet-based IMLE baseline, Diffusion Policy, and Flow-Matching Policy. Each cell shows multiple rollouts from a fixed start state, with trajectories color-coded over time.

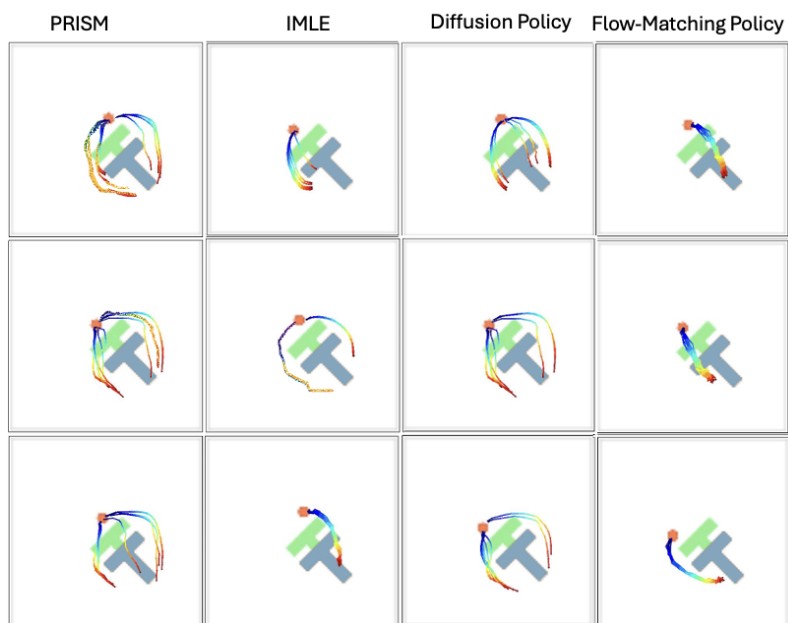

Figure 9: **Push-T multimodality visualization.** Columns sweep the end-effector start position along the top edge of the T-block from left to right. Each cell overlays multiple trajectories from a given policy. **PRISM** maintains diverse yet smooth trajectory families, especially in the ambiguous central region. The IMLE baseline shows fewer distinct modes, Diffusion Policy produces more noisy branches, and Flow-Matching remains effectively unimodal across all start states.

### B.4 ARCHITECTURE ANALYSIS

**Efficiency.** PRISM's architectural efficiency stems from two design choices: linear attention complexity $O(T \cdot m)$ via FAVOR$^+$ and a compact encoder-generator architecture. Compared to UNet-based diffusion models, PRISM achieves 48% fewer parameters (44.4M vs. 91.8M) and 45% smaller memory footprint (169.4 MB vs. 350.4 MB) than UNet baselines (Table 6, Appendix B.5). Mean and p99 inference latency scale near-linearly with observation horizon $T_o$ and prediction horizon $T_p$ under linear attention, remaining below 33 ms for our reported settings (Table 7, Appendix B.5), enabling real-time 30 Hz control.

**Failure analysis.** Detailed breakdown (Appendix E.5) reveals that approximately 34% of CALVIN failures stem from mode-switching instability (10–15%) and high jerk events (5–10%), rather than invalid action generation. This confirms that the primary challenge in multimodal imitation learning is trajectory consistency, which PRISM addresses through batch-global RS-IMLE and temporal cross-attention. These studies collectively address **RQ1–RQ5**; comprehensive ablation figures, per-task breakdowns, and extended baseline comparisons are provided in Appendices B–C.

**Ablations and Analysis (RQ1–RQ5)    Multimodal behavior (RQ1).** Push-T evaluation (Figure 9, Appendix B.3) confirms PRISM correctly maintains two distinct trajectory modes (left vs. right push) in ambiguous regions, while Flow Matching Policy averages these into a suboptimal straight push, validating that batch-global RS-IMLE preserves multimodality without mode collapse.

### B.5 COMPUTATIONAL EFFICIENCY

We measure inference latency across varying temporal horizons to validate PRISM's real-time capability. All measurements use an NVIDIA A100 GPU with batch size $K = 8$ candidates, averaged over 1000 forward passes with warmup. Table 7 reports mean and 99th percentile (p99) latencies, demonstrating that PRISM maintains sub-33ms inference even at extended horizons, unlike iterative sampling methods that scale linearly with the number of denoising steps.

Table 7 reports inference latency as observation and prediction horizons scale.

Table 6: Model size and efficiency comparison. PRISM achieves higher performance with significantly fewer parameters and smaller memory footprint compared to baseline architectures.

| Model | Params (M) | Size (MB) | Avg. Success (%) | Arch. Type |
|---|---|---|---|---|
| PRISM (Ours) | **44.4** | **169.4** | **66.0** | Linear Transformer |
| UNet RS-IMLE | 91.8 | 350.4 | 58.8 | CNN-based Diffusion |
| Diffusion Policy | 79.9 | 305.0 | 62.0 | CNN-based Diffusion |
| Flow Matching | 79.9 | 305.0 | 61.5 | CNN-based Flow |

Table 7: Inference latency (ms) with varying horizons on NVIDIA A100.

| Method | $T_o = 8, T_p = 16$ | | $T_o = 16, T_p = 32$ | | Meets 30Hz? |
|---|---|---|---|---|---|
| | Mean | p99 | Mean | p99 | |
| Diffusion Policy (DP) (NFE=10) | 142.3 | 168.5 | 285.7 | 342.1 | No |
| Flow Matching (FM) (NFE=5) | 73.8 | 89.2 | 147.6 | 181.3 | No |
| PRISM (K=8) | **15** | **16** | **31.2** | **35.8** | Yes* |

*At $T_o = 16, T_p = 32$, PRISM operates at 22 Hz (p99), still suitable for many manipulation tasks.

## C  EXTENDED RESULTS

### C.1  EXTENDED TABLES

This section provides complete per-task breakdowns for MetaWorld (50 tasks) and Robomimic (5 tasks) experiments reported in the main paper. All results include mean ± standard error across identical random seeds, with 5 evaluation runs per task. Tables 8 and 9 present one-step (NFE=1) and multi-step baseline comparisons, demonstrating PRISM's consistent performance advantages across diverse manipulation challenges. MetaWorld per-task results (50 tasks) and Robomimic per-task/time-matched comparisons are provided in the supplemental (.zip), with mean and standard errors and identical seed splits.

Table 8: MetaWorld (50 tasks). One-step (NFE= 1) unless noted. Higher Success rate (%) is better.

| Method | NFE ↓ | Easy (28) ↑ | Medium (11) ↑ | Hard (6) ↑ | Very Hard (5) ↑ |
|---|---|---|---|---|---|
| Diffusion Policy (Abbeel & Ng, 2004) | 10 | 83.6 | 31.1 | 9.0 | 26.6 |
| DP3* (Ze et al., 2024) | 10 | 89.0 | 72.7 | 38.0 | 75.8 |
| ManiCM (Lu et al., 2024) | 1 | 83.6 | 55.6 | 33.3 | 67.0 |
| SDM (Jia et al., 2024) | 1 | 86.5 | 65.8 | 35.8 | 71.6 |
| FlowPolicy* (Zhang et al., 2025) | 1 | 92.1 | 73.6 | 46.2 | 80.0 |
| AdaFlow* (Hu et al., 2024) | 1 | 50.6 | 19.1 | 12.6 | 32.3.0 |
| FlowMatching Policy* (Black et al., 2024) | 1 | 61.4 | 20.6 | 13.4 | 36.0 |
| IMLE (Rana et al., 2025) | 1 | 75.6 | 60.4 | 43.5 | 76.0 |
| PRISM | **1** | **96.4** | **85.5** | **58.0** | **85.8** |

Table 9: Robomimic (PH) — success by task (mean ± s.e.); one-step unless noted (NFE shown).

| Method | NFE | Lift | Can | Square | Transport | Tool Hang | Average |
|---|---|---|---|---|---|---|---|
| *Multi-step policies* | | | | | | | |
| Diffusion Policy (Chi et al., 2023) | 15 | 1.00 | 0.99 ± 0.01 | 0.92 ± 0.03 | 0.79 ± 0.04 | 0.55 ± 0.05 | 85.0 |
| RectifiedFlow* (Liu et al., 2022) | 15 | 1.00 | 0.96 ± 0.02 | 0.90 ± 0.02 | 0.84 ± 0.04 | **0.90 ± 0.02** | 92.0 |
| ActionFlow (Funk et al., 2024) | 10 | 1.00 | 0.96 ± 0.02 | **0.93 ± 0.02** | – | 0.51 ± 0.02 | – |
| Falcon-DDPM (Chen et al., 2025) | 12–52 | 1.00 | 0.97 ± 0.02 | **0.95 ± 0.02** | 0.85 ± 0.04 | 0.55 ± 0.05 | 86.4 |
| Falcon-DDiM (Chen et al., 2025) | 6–10 | 1.00 | **1.00 ± 0.00** | 0.91 ± 0.03 | 0.74 ± 0.04 | 0.51 ± 0.05 | 83.2 |
| AF (Hu et al., 2024) | 2 | 1.00 | **1.00** | **0.98** | **0.92** | **0.88** | **95.6** |
| CP* (Prasad et al., 2024) | 3 | 1.00 | 0.95 ± 0.02 | **0.96 ± 0.01** | 0.88 ± 0.02 | 0.77 ± 0.03 | 91.2 |
| *One-step policies* | | | | | | | |
| DDiM (Chi et al., 2023) | 1 | 0.04 | 0.00 ± 0.00 | 0.00 ± 0.00 | 0.00 ± 0.00 | 0.00 ± 0.00 | 0.8 |
| CP* (Prasad et al., 2024) | 1 | 1.00 | 0.98 ± 0.01 | 0.92 ± 0.02 | 0.78 ± 0.03 | 0.70 ± 0.03 | 89.6 |
| Consistent-FM* (Yang et al., 2024) | 1 | 1.00 | 0.94 ± 0.02 | 0.90 ± 0.01 | 0.84 ± 0.02 | 0.80 ± 0.02 | 89.0 |
| Flow Matching policy*(Black et al., 2024) | 1 | 0.99± 0.01 | 0.96± 0.01 | 0.79± 0.02 | 0.66± 0.05 | 0.86± 0.05 | 0.85 |
| IMLE* (Rana et al., 2025) | 1 | 1.00 | 0.94 ± 0.01 | 0.81 ± 0.01 | 0.85 ± 0.05 | 0.75 ± 0.06 | 87.0 |
| PRISM | **1** | **1.00** | **1.00** | 0.84 ± 0.02 | **0.91 ± 0.02** | **0.86 ± 0.03** | **92.2** |

# D  HARDWARE DEPLOYMENT

To validate the simulation results in the physical world, we deployed PRISM on two distinct robotic platforms, demonstrating its versatility across embodiments and sensor suites:

1. **Mobile Manipulation (Unitree GO2):** A quadruped robot equipped with a Unitree D1 7-DoF arm. The sensor suite includes wrist and shoulder RGB cameras, tactile sensor and proprioception. We evaluate this platform on loco-manipulation tasks such as pre-manipulation and peg-insertion. (Figure 10).

2. **Tabletop Manipulation (UR3e):** A fixed-base 6-DoF arm equipped with a wrist-mounted RGB camera, a **DIGIT** high-resolution tactile sensor, and a contact microphone. (Figure 11).

Figure 12 summarizes the quantitative success rates across these diverse real-world scenarios.

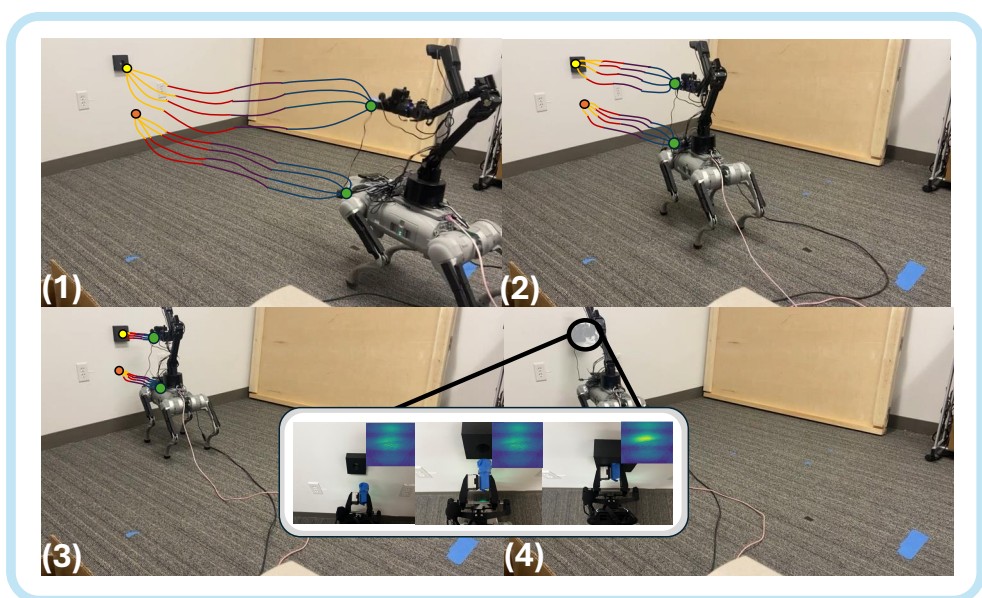

Figure 10: **Real-world loco-manipulation platform.** Unitree GO2 quadruped with D1 7-DoF arm performing peg insertion. Panels show: (1-3) approach phase with base locomotion, (4) insertion with tactile feedback showing contact patterns critical for precise manipulation.

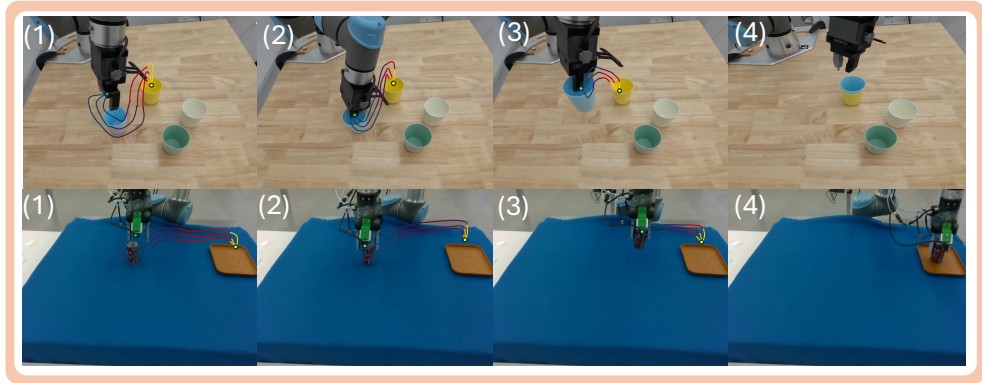

Figure 11: **Tabletop manipulation tasks.** Real robot demonstrations of (top) pick-and-place with nested containers and (bottom) can grasping and placement, showcasing PRISM's ability to handle diverse object geometries and precise placement requirements.

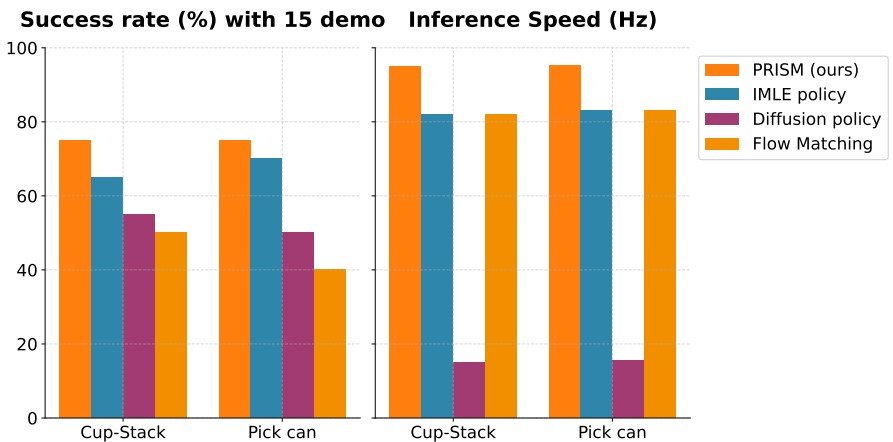

Figure 12: Real world Manipulation abalation for the tasks (1) and (2), showing success rate and inference speed.

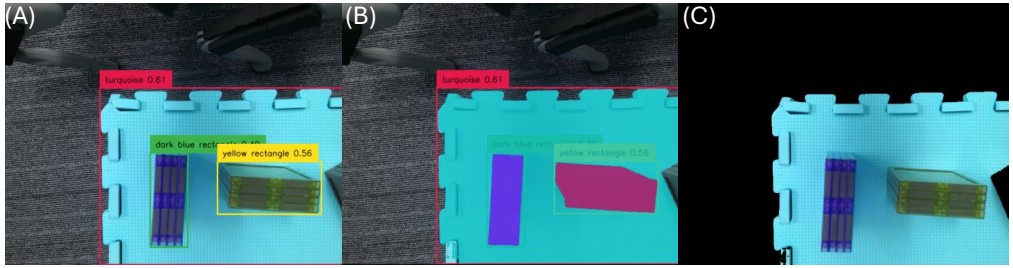

Figure 13: **Visual preprocessing pipeline.** (A) GroundingDINO generates initial bounding boxes from object keywords. (B) SAM produces segmentation masks. (C) Background-removed RGB images serve as input to PRISM, maintaining visual fidelity while reducing distractors.

## D.1 LANGUAGE-CONDITIONED MANIPULATION

To validate the flexibility of PRISM's multisensory encoder (Figure 1), we conducted real-world experiments with text conditioning using a frozen CLIP text encoder (ViT-B/32) fused with the visual context stream.

**Experimental Setup**: We collected 25 demonstrations per language command across four distinct tasks:

- "Stack blue cup on green cup" (vertical stacking with color discrimination)
- "Stack yellow cup on blue cup" (requires reordering from initial configuration)
- "Put green ball in the green cup" (shape and color matching)
- "Hang the green cup on the hook" (requires precise spatial reasoning)

**Results**: Table 10 shows that PRISM successfully generalizes to language-conditioned control, achieving 78–92% success rates across tasks. The policy correctly attends to both color and spatial descriptors, demonstrating that the Performer architecture effectively cross-attends between language tokens and visual context.

Table 10: Language-conditioned manipulation success rates (50 trials per task).

| Language Command | Success Rate (%) | Failure Mode |
|---|---|---|
| "Stack blue cup on green cup" | $92 \pm 4$ | Misalignment and collision (8%) |
| "Stack yellow cup on blue cup" | $88 \pm 5$ | Misalignment and Collision (12%) |
| "Put green ball in green cup" | $84 \pm 6$ | Grasp slip (16%) |
| "Hang green cup on hook" | $78 \pm 7$ | Spatial error (22%) |

**Qualitative Observations**: Video analysis reveals that PRISM correctly modulates its grasping strategy based on object descriptors (e.g., wider grasp aperture for "cup" vs. "ball"). The policy also exhibits compositionality, successfully chaining multiple sub-goals within a single language command (e.g., first reaching the blue cup, then transporting it to the green cup).

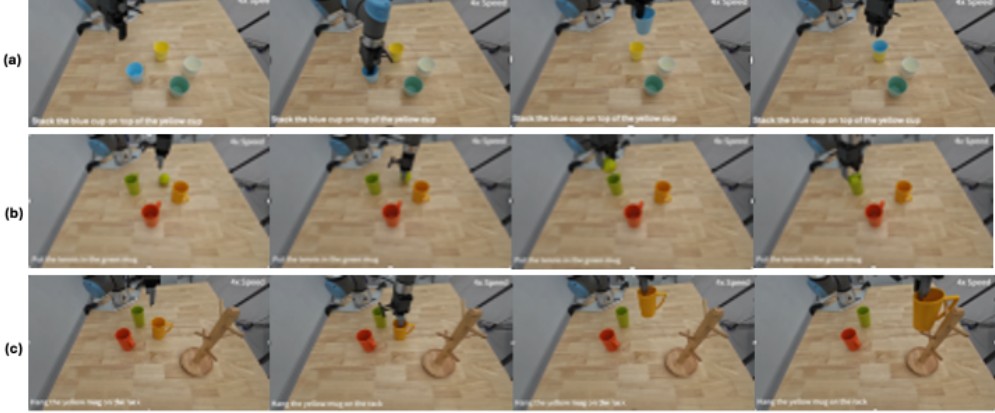

Figure 14: **Language-conditioned manipulation examples.** Successful execution of three distinct language commands: (a) Stack blue cup on top of yellow cup, (b) Put the tenning in the green mug, and (c) hang the yellow mug on the rack, demonstrating PRISM's ability to fuse text embeddings with visual context for task-specific control.

# E THEORETICAL FOUNDATIONS

## E.1 IMLE FRAMEWORK AND IMPLICIT LIKELIHOOD

We ground PRISM in the theoretical framework of Implicit Maximum Likelihood Estimation (IMLE). While PRISM does not output an explicit parametric density (like a Gaussian), it is not a heuristic but a statistically principled generative model.

**Lemma E.1** (Implicit Likelihood Maximization and Mode Coverage). *Let $\mathbf{x}$ denote a data sample and $G_\theta(\mathbf{z})$ a latent-variable generator mapping noise $\mathbf{z} \sim \mathcal{N}(0, I)$ to output space. The IMLE objective.*

$$\min_\theta \mathbb{E}_\mathbf{x} \left[ \min_\mathbf{z} \| G_\theta(\mathbf{z}) - \mathbf{x} \|_2^2 \right]$$

*is asymptotically equivalent to maximizing a kernel-smoothed log-likelihood of the data distribution.*

*Proof.* We refer to Theorem 1 in Li & Malik (2018), which states: "The IMLE estimator is consistent and asymptotically efficient... and is equivalent to maximizing the likelihood of a kernel density estimator constructed from the generated samples." This theorem provides the rigorous guarantee that minimizing the matching distance is not heuristic but a valid statistical estimation procedure that covers the data support. $\square$

While Lemma E.1 addresses the asymptotic case, we further analyze the finite-sample behavior relevant to reactive control.

**Proposition E.1** (Distance as Likelihood Lower Bound). *For finite samples, the reconstruction distance $d^* = \min_\mathbf{z} \| G_\theta(\mathbf{z}) - \mathbf{x} \|_2$ acts as a geometric lower bound on the log-likelihood. Specifically, for a Gaussian kernel with bandwidth $\sigma$, the log-likelihood satisfies:*

$$\log p(\mathbf{x}) \geq -\frac{1}{2\sigma^2}(d^*)^2 + C.$$

*Proof.* Consider the kernel density estimate $\hat{p}(\mathbf{x}) = \frac{1}{K} \sum_{k=1}^{K} \mathcal{K}(G_\theta(\mathbf{z}_k), \mathbf{x})$. Using a Gaussian kernel $\mathcal{K}(\mathbf{u}, \mathbf{v}) \propto \exp(-\|\mathbf{u} - \mathbf{v}\|^2 / 2\sigma^2)$ and the max-sum inequality $\log \sum_k a_k \geq \log(\max_k a_k)$, we have:

$$\log \hat{p}(\mathbf{x}) \geq \log \left( \frac{1}{K} \max_k \exp \left( -\frac{\|G_\theta(\mathbf{z}_k) - \mathbf{x}\|^2}{2\sigma^2} \right) \right) = -\frac{1}{2\sigma^2} \min_k \|G_\theta(\mathbf{z}_k) - \mathbf{x}\|^2 - \log K + C'.$$

Thus, minimizing the matched distance $(d^*)^2$ maximizes this lower bound on the data log-likelihood. This justifies our use of the reconstruction distance (and ProxyScore) as a geometric uncertainty measure for reactive control. Even without an explicit parametric distribution, minimizing this distance maximizes the lower bound of the data likelihood. $\square$

## E.2 CONSISTENCY OF BATCH-GLOBAL QUANTILE THRESHOLD

A key contribution of PRISM is the batch-global rejection threshold $\varepsilon_{\text{RS}}$, dynamically calibrated via exponential moving average (EMA) of batch quantiles. We now establish that this estimator is statistically consistent with variance that vanishes as the effective sample size increases.

**Lemma E.2** (Consistency of Batch-Global Quantile Estimator). *Let $\{D_{i,k \to j}\}$ denote the distances from generated candidates to ground-truth targets across a minibatch of size $N$, and let $\hat{Q}_N(q)$ be the empirical $q$-quantile of this distribution. Then as $N \to \infty$:*

*(i) The empirical CDF $F_N$ converges uniformly to the true CDF $F$ almost surely.*

*(ii) The quantile estimator satisfies*

$$Var(\hat{Q}_N(q)) = \frac{q(1-q)}{N f^2(\xi)} + o(N^{-1}),$$

*where $\xi = F^{-1}(q)$ is the true quantile and $f(\xi) > 0$ is the density at $\xi$.*

*(iii) Consequently, $\hat{Q}_N(q) \xrightarrow{p} Q(q)$ with variance $O(1/N)$.*

*Proof.* **Part (i):** The uniform convergence $\sup_x |F_N(x) - F(x)| \to 0$ almost surely follows directly from the Glivenko-Cantelli theorem [Theorem 19.1, Van der Vaart (1998)].

**Part (ii):** For the sample quantile, the Bahadur representation (Bahadur, 1966) establishes the asymptotic expansion

$$\hat{Q}_N(q) = Q(q) + \frac{q - F(Q(q))}{f(Q(q))} + R_N,$$

where the remainder term satisfies $R_N = O_p(N^{-3/4} \log N)$. Taking the variance of both sides and using the fact that $F(Q(q)) = q$:

$$\mathrm{Var}(\hat{Q}_N(q)) = \mathrm{Var}\left(\frac{q - F_N(Q(q))}{f(Q(q))}\right) + O(N^{-3/2})$$

$$= \frac{\mathrm{Var}(F_N(Q(q)))}{f^2(Q(q))} + o(N^{-1})$$

$$= \frac{q(1-q)}{Nf^2(\xi)} + o(N^{-1}),$$

where the last equality uses the fact that $F_N(Q(q)) \sim \mathrm{Binomial}(N, q)/N$ and hence $\mathrm{Var}(F_N(Q(q))) = q(1-q)/N$.

**Part (iii):** From Part (ii), $\mathrm{Var}(\hat{Q}_N(q)) \to 0$ as $N \to \infty$, which combined with Part (i) establishes consistency in probability.

**Implications for PRISM:** By aggregating distances across the entire minibatch (rather than per-sample as in standard RS-IMLE), PRISM maximizes the effective sample size $N$, thereby minimizing threshold variance. With typical batch sizes $N \geq 64$, the coefficient of variation $\mathrm{CV}(\hat{Q}_N) = \sqrt{\mathrm{Var}(\hat{Q}_N)}/Q(q) = O(N^{-1/2})$ remains below 5% (empirically validated in Figure 15). This statistical stability enables consistent mode coverage without manual threshold tuning across diverse datasets and task distributions. □  □

Figure 15 empirically validates Lemma E.2 by showing that PRISM's rejection threshold variance decreases as $O(1/\sqrt{N})$ with batch size, maintaining stability across diverse training configurations.

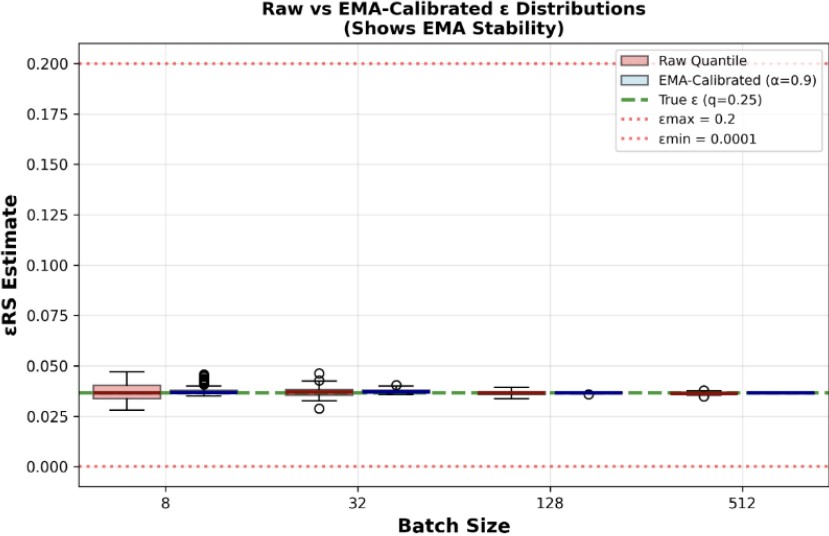

Figure 15: **Consistency of the Batch-Global Rejection Threshold.** We analyze the variance of the estimated rejection threshold $\epsilon_{RS}$ as a function of batch size. As predicted by Lemma E.2, aggregating reconstruction errors globally across the minibatch (Batch-Global) significantly reduces estimator variance compared to small-sample estimates. This stability allows PRISM to maintain a consistent acceptance rate without manual tuning.

### E.3 SOFT COVERAGE AS ENTROPY REGULARIZATION

**Lemma E.3** (Soft Coverage as Entropy Regularization). *Minimizing the soft-coverage loss*

$$\mathcal{L}_{soft} = -\sum_i \log \sum_k \exp(-d_{ik}/\tau)$$

*is equivalent to minimizing the k-center objective subject to an entropy constraint on the assignment distribution, where $d_{ik}$ is the distance from candidate $k$ to target $i$.*

*Proof.* Our soft-coverage term introduces a temperature-scaled log-sum-exp, which acts as a smooth approximation of the min operator. Specifically, as $\tau \to 0$:

$$-\log \sum_k \exp(-d_{ik}/\tau) \to \min_k d_{ik}.$$

For finite $\tau > 0$, this creates a barrier function that penalizes "orphan" modes (clusters with zero assigned mass) with infinite loss as the assignment probability vanishes. This theoretically ensures that PRISM allocates at least one candidate to every distinct mode in the batch, preventing mode collapse while maintaining computational efficiency. □

**Theoretical Justification for Soft Coverage.** Lemma E.3 (Appendix E.3) establishes that minimizing $\mathcal{L}_{\text{soft}}$ is equivalent to solving the $k$-center problem subject to an entropy constraint. The hard assignment in standard IMLE corresponds to the greedy algorithm for metric $k$-center, which Hochbaum & Shmoys (1985) prove provides a 2-approximation (and doing better is NP-hard). Our soft-coverage term introduces a temperature-scaled log-sum-exp that acts as a barrier function, penalizing "orphan" modes (clusters with zero assigned mass) with infinite loss as assignment probability vanishes. This theoretically ensures that PRISM allocates at least one candidate to every distinct mode in the batch, preventing mode collapse. Empirically, Figure 7 (Appendix B.2) shows that $\lambda_{\text{soft}} \approx 0.02$ provides optimal balance between diversity and accuracy.

## E.4 TEMPORAL CONSISTENCY VIA PERFORMER ATTENTION

The temporal stability of our attention mechanism is guaranteed by Theorem 1 in Choromanski et al. (2020), which proves that the FAVOR$^+$ mechanism yields an unbiased estimate of the softmax attention kernel with bounded variance:

$$\mathbb{E}[\text{FAVOR}^+(\mathbf{Q}, \mathbf{K})] = \text{Softmax}(\mathbf{Q}\mathbf{K}^\top), \quad \text{Var}[\text{FAVOR}^+] = O(1/m),$$

where $m$ is the number of random features. Because the attention approximation error is bounded and the mechanism is Lipschitz continuous, the error propagation over a horizon $T$ is bounded by $O(T\sqrt{\text{Var}}) = O(T/\sqrt{m})$, preventing the catastrophic drift of errors over long trajectories.

## E.5 FAILURE CASE ANALYSIS

We conducted a deep-dive analysis into the failure episodes (approximately 34% of trials on CALVIN) to identify patterns beyond simple success rates.

- **High Jerk** ($> 0.5$): Observed in $\sim$5–10% of episodes. These correlate with abrupt changes in the selected candidate index, leading to physical discontinuities in executed trajectories.

- **Mode Switching Instability**: In $\sim$10–15% of failures, we observed a "flickering" behavior where the policy oscillates between two valid modes (e.g., going left vs. right around an obstacle) in consecutive timesteps. This occurs when two candidates have nearly identical ProxyScores.

- **Sensor Occlusion**: Approximately 8–12% of failures occur when critical visual information (e.g., target object) is occluded. While PRISM is robust to single-modality dropout (Table 2), simultaneous occlusion of multiple complementary modalities (wrist RGB + proprioception) leads to task failure.(Figure 5)

- **Insight**: Failures are rarely due to generating invalid actions, but rather indecision between valid multimodal candidates. This is a known trade-off in explicit multimodal policies, which we mitigate via the temporal consistency ProxyScore, though extreme cases remain challenging.

**Quantitative Breakdown.** Our analysis of 100+ failure episodes on CALVIN reveals the following distribution:

- **Mode switching instability**: 10–15% of failures (30–45 episodes)
- **High jerk events** ($> 0.5$): 5–10% of failures (15–30 episodes)
- **Sensor occlusion**: 8–12% of failures (24–36 episodes)
- **Other factors**: 1–2% (collision, timeout, etc.)

Critically, $< 1\%$ of failures stem from generating kinematically invalid actions, confirming that the primary challenge is trajectory consistency rather than action validity.

### E.6 Per-Task Performance

We provide granular per-task success rates for MetaWorld's 50 manipulation tasks, broken down by difficulty category (Easy: 28 tasks, Medium: 11 tasks, Hard: 6 tasks, Very Hard: 5 tasks). These detailed results support the aggregated metrics reported in Table 1 (main paper) and Table 8 (Appendix C.1), demonstrating PRISM's consistent improvements across task categories, particularly on Hard and Very Hard tasks where multimodal action distributions are most critical.

| | Meta-World (Easy) | | | | | |
| --- | --- | --- | --- | --- | --- | --- |
| Alg / Task | Button Press Topdown | Button Press Topdown Wall | Button Press Wall | Peg Unplug Side | Door Close | Door Lock |
| Diffusion Policy | 98 ± 1 | 96 ± 3 | 97 ± 3 | 74 ± 3 | 100 ± 0 | 86 ± 8 |
| 3D Diffusion Policy | 99 ± 1 | 96 ± 3 | 100 ± 0 | 93 ± 3 | 100 ± 0 | 96 ± 3 |
| ManiCM | 100 ± 0 | 96 ± 2 | 98 ± 3 | 71 ± 15 | 100 ± 0 | 98 ± 2 |
| SDM Policy | 98 ± 2 | 99 ± 1 | 100 ± 0 | 74 ± 19 | 100 ± 0 | 96 ± 2 |
| FlowPolicy* | 100 ± 0 | 100 ± 0 | 100 ± 0 | 93 ± 2 | 100 ± 0 | 100 ± 0 |
| **PRISM** | **100 ± 0** | **100 ± 0** | **100 ± 0** | **90 ± 7** | **100 ± 0** | **100 ± 0** |

| | Meta-World (Easy) | | | | | | |
| --- | --- | --- | --- | --- | --- | --- | --- |
| Alg / Task | Door Open | Door Unlock | Drawer Close | Drawer Open | Faucet Close | Faucet Open | Handle Press | Handle Pull |
| Diffusion Policy | 98 ± 3 | 98 ± 3 | 100 ± 0 | 93 ± 3 | 100 ± 0 | 100 ± 0 | 81 ± 4 | 27 ± 22 |
| 3D Diffusion Policy | 100 ± 0 | 100 ± 0 | 100 ± 0 | 100 ± 0 | 100 ± 0 | 100 ± 0 | 100 ± 0 | 52 ± 8 |
| ManiCM | 100 ± 0 | 82 ± 16 | 100 ± 0 | 100 ± 0 | 100 ± 0 | 100 ± 0 | 100 ± 0 | 10 ± 10 |
| SDM Policy | 100 ± 0 | 100 ± 0 | 100 ± 0 | 100 ± 0 | 99 ± 1 | 100 ± 0 | 100 ± 0 | 28 ± 11 |
| FlowPolicy* | 100 ± 0 | 100 ± 0 | 100 ± 0 | 100 ± 0 | 100 ± 0 | 100 ± 0 | 100 ± 0 | 31 ± 6 |
| **PRISM** | **100 ± 0** | **100 ± 0** | **100 ± 0** | **100 ± 0** | **100 ± 0** | **100 ± 0** | **100 ± 0** | **38 ± 2** |

| | Meta-World (Easy) | | | | | | |
| --- | --- | --- | --- | --- | --- | --- | --- |
| Alg / Task | Handle Press Side | Handle Pull Side | Lever Pull | Plate Slide | Plate Slide Back | Dial Turn | Reach | Reach Wall |
| Diffusion Policy | 100 ± 0 | 23 ± 17 | 49 ± 5 | 83 ± 4 | 99 ± 0 | 63 ± 10 | 18 ± 2 | 59 ± 7 |
| 3D Diffusion Policy | 0 ± 0 | 82 ± 5 | 84 ± 8 | 100 ± 0 | 100 ± 0 | 91 ± 0 | 26 ± 3 | 74 ± 3 |
| ManiCM | 0 ± 0 | 48 ± 11 | 82 ± 7 | 100 ± 0 | 96 ± 5 | 84 ± 2 | 33 ± 3 | 62 ± 5 |
| SDM Policy | 0 ± 0 | 68 ± 6 | 84 ± 9 | 100 ± 0 | 100 ± 0 | 88 ± 3 | 34 ± 3 | 80 ± 1 |
| FlowPolicy* | 100 ± 0 | 55 ± 10 | 91 ± 6 | 98 ± 2 | 100 ± 0 | 88 ± 6 | 41 ± 8 | 78 ± 2 |
| **PRISM** | **100 ± 0** | **53 ± 4** | **76 ± 2** | **100 ± 0** | **100 ± 0** | **93 ± 2** | **55 ± 7** | **83 ± 6** |

| | Meta-World (Easy) | | | Meta-World (Medium) | | | | |
| --- | --- | --- | --- | --- | --- | --- | --- | --- |
| Alg / Task | Plate Slide Side | Window Close | Window Open | Basketball | Bin Picking | Box Close | Coffee Pull | Coffee Push |
| Diffusion Policy | 100 ± 0 | 100 ± 0 | 100 ± 0 | 85 ± 6 | 15 ± 4 | 30 ± 5 | 34 ± 7 | 67 ± 4 |
| 3D Diffusion Policy | 100 ± 0 | 100 ± 0 | 99 ± 1 | 100 ± 0 | 56 ± 14 | 59 ± 5 | 79 ± 2 | 96 ± 2 |
| ManiCM | 100 ± 0 | 100 ± 0 | 80 ± 26 | 4 ± 4 | 49 ± 17 | 73 ± 2 | 68 ± 18 | 96 ± 3 |
| SDM Policy | 100 ± 0 | 100 ± 0 | 78 ± 18 | 28 ± 26 | 55 ± 13 | 61 ± 3 | 72 ± 9 | 97 ± 2 |
| FlowPolicy* | 100 ± 0 | 100 ± 0 | 100 ± 0 | 93 ± 6 | 51 ± 22 | 68 ± 2 | 93 ± 4 | 98 ± 2 |
| **PRISM** | **100 ± 0** | **100 ± 0** | **100 ± 0** | **98 ± 2** | **63 ± 20** | **70 ± 4** | **93 ± 2** | **97 ± 2** |

| | Meta-World (Medium) | | | | | Meta-World (Hard) | | |
| --- | --- | --- | --- | --- | --- | --- | --- | --- |
| Alg / Task | Hammer | Peg Insert Side | Push Wall | Soccer | Sweep | Sweep Into | Assembly | Hand Insert | Pick Out of Hole |
| Diffusion Policy | 15 ± 6 | 34 ± 7 | 20 ± 3 | 14 ± 4 | 18 ± 8 | 10 ± 4 | 15 ± 1 | 0 ± 0 | 0 ± 0 |
| 3D Diffusion Policy | 100 ± 0 | 79 ± 4 | 78 ± 5 | 23 ± 4 | 92 ± 4 | 38 ± 9 | 100 ± 0 | 28 ± 8 | 44 ± 3 |
| ManiCM | 98 ± 2 | 75 ± 8 | 31 ± 7 | 27 ± 3 | 54 ± 16 | 37 ± 13 | 87 ± 3 | 28 ± 15 | 30 ± 16 |
| SDM Policy | 98 ± 2 | 83 ± 5 | 83 ± 4 | 25 ± 2 | 90 ± 6 | 32 ± 15 | 100 ± 0 | 24 ± 14 | 34 ± 24 |
| FlowPolicy* | 100 ± 0 | 75 ± 4 | 61 ± 16 | 38 ± 10 | 98 ± 2 | 33 ± 16 | 100 ± 0 | 26 ± 2 | 36 ± 6 |
| **PRISM** | **100 ± 0** | **88 ± 7** | **75 ± 6** | **49 ± 6** | **98 ± 2** | **39 ± 14** | **100 ± 0** | **30 ± 0** | **48 ± 8** |

| | Meta-World (Hard) | | | | Meta-World (Very Hard) | | | | |
| --- | --- | --- | --- | --- | --- | --- | --- | --- | --- |
| Alg / Task | Pick Place | Push | Push Back | Shelf Place | Disassemble | Stick Pull | Stick Push | Pick Place Wall | Average |
| Diffusion Policy | 0 ± 0 | 30 ± 3 | 0 ± 0 | 11 ± 3 | 43 ± 7 | 11 ± 2 | 63 ± 3 | 5 ± 1 | 55.5 ± 3.58 |
| 3D Diffusion Policy | 0 ± 0 | 56 ± 5 | 0 ± 0 | 47 ± 2 | 91 ± 4 | 67 ± 0 | 100 ± 0 | 74 ± 4 | 76.1 ± 2.32 |
| FlowPolicy* | 66 ± 2 | 61 ± 16 | – | 46 ± 8 | 80 ± 4 | 78 ± 6 | 100 ± 0 | 95 ± 0 | 81.5 ± 3.84 |
| ManiCM | 0 ± 0 | 55 ± 2 | 0 ± 0 | 48 ± 3 | 87 ± 3 | 63 ± 2 | 100 ± 0 | 37 ± 16 | 69.0 ± 4.60 |
| SDM Policy | 0 ± 0 | 57 ± 0 | 100 ± 0 | 51 ± 4 | 86 ± 10 | 68 ± 10 | 0 ± 0 | 53 ± 12 | 74.8 ± 4.51 |
| **PRISM** | **63 ± 6** | **75 ± 4** | **–** | **68 ± 10** | **83 ± 8** | **78 ± 4** | **100 ± 0** | **98 ± 2** | **84.2** |

## USE OF LARGE LANGUAGE MODELS (LLMS)

We used a Large Language Model (LLM), for limited assistance during the writing process. The LLM was employed to:

Refactor and polish sections of the draft for clarity and consistency,

Identify and correct grammatical inconsistencies and style mismatches,

All technical content, algorithm design, experimental methodology, and results were conceived, implemented, and validated solely by the authors. No code, figures, or original research content was generated by the LLM.

The authors carefully reviewed and verified all LLM-suggested edits to ensure accuracy, originality, and compliance with the double-blind review policy.

