| $98 \pm 1$ | $96 \pm 3$ | $97 \pm 3$ | $74 \pm 3$ | $100 \pm 0$ | $86 \pm 8$ |
| 3D Diffusion Policy | $99 \pm 1$ | $96 \pm 3$ | $100 \pm 0$ | $93 \pm 3$ | $100 \pm 0$ | $96 \pm 3$ |
| ManiCM | $100 \pm 0$ | $96 \pm 2$ | $98 \pm 3$ | $71 \pm 15$ | $100 \pm 0$ | $98 \pm 2$ |
| SDM Policy | $98 \pm 2$ | $99 \pm 1$ | $100 \pm 0$ | $74 \pm 19$ | $100 \pm 0$ | $96 \pm 2$ |
| FlowPolicy* | $100 \pm 0$ | $100 \pm 0$ | $100 \pm 0$ | $93 \pm 2$ | $100 \pm 0$ | $100 \pm 0$ |
| **PRISM** | **$100 \pm 0$** | **$100 \pm 0$** | **$100 \pm 0$** | **$90 \pm 7$** | **$100 \pm 0$** | **$100 \pm 0$** |

| | Meta-World (Easy) | | | | | | |
|---|---|---|---|---|---|---|---|
| Alg / Task | Door Open | Door Unlock | Drawer Close | Drawer Open | Faucet Close | Faucet Open | Handle Press | Handle Pull |
| Diffusion Policy | $98 \pm 3$ | $98 \pm 3$ | $100 \pm 0$ | $93 \pm 3$ | $100 \pm 0$ | $100 \pm 0$ | $81 \pm 4$ | $27 \pm 22$ |
| 3D Diffusion Policy | $100 \pm 0$ | $100 \pm 0$ | $100 \pm 0$ | $100 \pm 0$ | $100 \pm 0$ | $100 \pm 0$ | $100 \pm 0$ | $52 \pm 8$ |
| ManiCM | $100 \pm 0$ | $82 \pm 16$ | $100 \pm 0$ | $100 \pm 0$ | $100 \pm 0$ | $100 \pm 0$ | $100 \pm 0$ | $10 \pm 10$ |
| SDM Policy | $100 \pm 0$ | $100 \pm 0$ | $100 \pm 0$ | $100 \pm 0$ | $99 \pm 1$ | $100 \pm 0$ | $100 \pm 0$ | $28 \pm 11$ |
| FlowPolicy* | $100 \pm 0$ | $100 \pm 0$ | $100 \pm 0$ | $100 \pm 0$ | $100 \pm 0$ | $100 \pm 0$ | $100 \pm 0$ | $31 \pm 6$ |
| **PRISM** | **$100 \pm 0$** | **$100 \pm 0$** | **$100 \pm 0$** | **$100 \pm 0$** | **$100 \pm 0$** | **$100 \pm 0$** | **$100 \pm 0$** | **$38 \pm 2$** |

| | Meta-World (Easy) | | | | | | |
|---|---|---|---|---|---|---|---|
| Alg / Task | Handle Press Side | Handle Pull Side | Lever Pull | Plate Slide | Plate Slide Back | Dial Turn | Reach | Reach Wall |
| Diffusion Policy | $100 \pm 0$ | $23 \pm 17$ | $49 \pm 5$ | $83 \pm 4$ | $99 \pm 0$ | $63 \pm 10$ | $18 \pm 2$ | $59 \pm 7$ |
| 3D Diffusion Policy | $0 \pm 0$ | $82 \pm 5$ | $84 \pm 8$ | $100 \pm 0$ | $100 \pm 0$ | $91 \pm 0$ | $26 \pm 3$ | $74 \pm 3$ |
| ManiCM | $0 \pm 0$ | $48 \pm 11$ | $82 \pm 7$ | $100 \pm 0$ | $96 \pm 5$ | $84 \pm 2$ | $33 \pm 3$ | $62 \pm 5$ |
| SDM Policy | $0 \pm 0$ | $68 \pm 6$ | $84 \pm 9$ | $100 \pm 0$ | $100 \pm 0$ | $88 \pm 3$ | $34 \pm 3$ | $80 \pm 1$ |
| FlowPolicy* | $100 \pm 0$ | $55 \pm 10$ | $91 \pm 6$ | $98 \pm 2$ | $100 \pm 0$ | $88 \pm 6$ | $41 \pm 8$ | $78 \pm 2$ |
| **PRISM** | **$100 \pm 0$** | **$53 \pm 4$** | **$76 \pm 2$** | **$100 \pm 0$** | **$100 \pm 0$** | **$93 \pm 2$** | **$55 \pm 7$** | **$83 \pm 6$** |

| | Meta-World (Easy) | | | Meta-World (Medium) | | | | |
|---|---|---|---|---|---|---|---|---|
| Alg / Task | Plate Slide Side | Window Close | Window Open | Basketball | Bin Picking | Box Close | Coffee Pull | Coffee Push |
| Diffusion Policy | $100 \pm 0$ | $100 \pm 0$ | $100 \pm 0$ | $85 \pm 6$ | $15 \pm 4$ | $30 \pm 5$ | $34 \pm 7$ | $67 \pm 4$ |
| 3D Diffusion Policy | $100 \pm 0$ | $100 \pm 0$ | $99 \pm 1$ | $100 \pm 0$ | $56 \pm 14$ | $59 \pm 5$ | $79 \pm 2$ | $96 \pm 2$ |
| ManiCM | $100 \pm 0$ | $100 \pm 0$ | $80 \pm 26$ | $4 \pm 4$ | $49 \pm 17$ | $73 \pm 2$ | $68 \pm 18$ | $96 \pm 3$ |
| SDM Policy | $100 \pm 0$ | $100 \pm 0$ | $78 \pm 18$ | $28 \pm 26$ | $55 \pm 13$ | $61 \pm 3$ | $72 \pm 9$ | $97 \pm 2$ |
| FlowPolicy* | $100 \pm 0$ | $100 \pm 0$ | $100 \pm 0$ | $93 \pm 6$ | $51 \pm 22$ | $68 \pm 2$ | $93 \pm 4$ | $98 \pm 2$ |
| **PRISM** | **$100 \pm 0$** | **$100 \pm 0$** | **$100 \pm 0$** | **$98 \pm 2$** | **$63 \pm 20$** | **$70 \pm 4$** | **$93 \pm 2$** | **$97 \pm 2$** |

| | Meta-World (Medium) | | | | | Meta-World (Hard) | | |
|---|---|---|---|---|---|---|---|---|
| Alg / Task | Hammer | Peg Insert Side | Push Wall | Soccer | Sweep | Sweep Into | Assembly | Hand Insert | Pick Out of Hole |
| Diffusion Policy | $15 \pm 6$ | $34 \pm 7$ | $20 \pm 3$ | $14 \pm 4$ | $18 \pm 8$ | $10 \pm 4$ | $15 \pm 1$ | $0 \pm 0$ | $0 \pm 0$ |
| 3D Diffusion Policy | $100 \pm 0$ | $79 \pm 4$ | $78 \pm 5$ | $23 \pm 4$ | $92 \pm 4$ | $38 \pm 9$ | $100 \pm 0$ | $28 \pm 8$ | $44 \pm 3$ |
| ManiCM | $98 \pm 2$ | $75 \pm 8$ | $31 \pm 7$ | $27 \pm 3$ | $54 \pm 16$ | $37 \pm 13$ | $87 \pm 3$ | $28 \pm 15$ | $30 \pm 16$ |
| SDM Policy | $98 \pm 2$ | $83 \pm 5$ | $83 \pm 4$ | $25 \pm 2$ | $90 \pm 6$ | $32 \pm 15$ | $100 \pm 0$ | $24 \pm 14$ | $34 \pm 24$ |
| FlowPolicy* | $100 \pm 0$ | $75 \pm 4$ | $61 \pm 16$ | $38 \pm 10$ | $98 \pm 2$ | $33 \pm 16$ | $100 \pm 0$ | $26 \pm 2$ | $36 \pm 6$ |
| **PRISM** | **$100 \pm 0$** | **$88 \pm 7$** | **$75 \pm 6$** | **$49 \pm 6$** | **$98 \pm 2$** | **$39 \pm 14$** | **$100 \pm 0$** | **$30 \pm 0$** | **$48 \pm 8$** |

| | Meta-World (Hard) | | | | Meta-World (Very Hard) | | | | |
|---|---|---|---|---|---|---|---|---|---|
| Alg / Task | Pick Place | Push | Push Back | Shelf Place | Disassemble | Stick Pull | Stick Push | Pick Place Wall | Average |
| Diffusion Policy | $0 \pm 0$ | $30 \pm 3$ | $0 \pm 0$ | $11 \pm 3$ | $43 \pm 7$ | $11 \pm 2$ | $63 \pm 3$ | $5 \pm 1$ | $55.5 \pm 3.58$ |
| 3D Diffusion Policy | $0 \pm 0$ | $56 \pm 5$ | $0 \pm 0$ | $47 \pm 2$ | $91 \pm 4$ | $67 \pm 0$ | $100 \pm 0$ | $74 \pm 4$ | $76.1 \pm 2.32$ |
| FlowPolicy* | $66 \pm 2$ | $61 \pm 16$ | – | $46 \pm 8$ | $80 \pm 4$ | $78 \pm 6$ | $100 \pm 0$ | $95 \pm 0$ | $81.5 \pm 3.84$ |
| ManiCM | $0 \pm 0$ | $55 \pm 2$ | $0 \pm 0$ | $48 \pm 3$ | $87 \pm 3$ | $63 \pm 2$ | $100 \pm 0$ | $37 \pm 16$ | $69.0 \pm 4.60$ |
| SDM Policy | $0 \pm 0$ | $57 \pm 0$ | $100 \pm 0$ | $51 \pm 4$ | $86 \pm 10$ | $68 \pm 10$ | $0 \pm 0$ | $53 \pm 12$ | $74.8 \pm 4.51$ |
| **PRISM** | **$63 \pm 6$** | **$75 \pm 4$** | **–** | **$68 \pm 10$** | **$83 \pm 8$** | **$78 \pm 4$** | **$100 \pm 0$** | **$98 \pm 2$** | **$84.2$** |

## USE OF LARGE LANGUAGE MODELS (LLMS)

We used a Large Language Model (LLM), for limited assistance during the writing process. The LLM was employed to:

Refactor and polish sections of the draft for clarity and consistency,

Identify and correct grammatical inconsistencies and style mismatches,

All technical content, algorithm design, experimental methodology, and results were conceived, implemented, and validated solely by the authors. No code, figures, or original research content was generated by the LLM.

The authors carefully reviewed and verified all LLM-suggested edits to ensure accuracy, originality, and compliance with the double-blind review policy.