# OpenReview forum: "PRISM: Performer RS-IMLE for Single-pass Multisensory Imitation Learning"
_ICLR.cc/2026/Conference — ICLR 2026 Conference Desk Rejected Submission_

### Official Review · Reviewer_FNZA · 2025-10-30

**Soundness:** 3
**Presentation:** 1
**Contribution:** 2
**Rating:** 2
**Confidence:** 2

**Summary:**

PRISM introduces a fast, single-pass imitation learning policy that handles rich sensory streams (vision, depth, tactile, proprioception, and audio when available) and produces full action sequences in one forward computation. Instead of relying on slow iterative sampling like diffusion or flow-matching models, PRISM combines a temporal multisensory encoder with a Performer-based linear-attention generator and augments IMLE training with a batch-global rejection mechanism to maintain multimodal action diversity and trajectory smoothness. The policy generates multiple candidate futures in parallel, selects a coherent action sequence in a receding-horizon manner, and achieves real-time control rates with significantly smoother motions. Experiments across MetaWorld, CALVIN, Robomimic, and Unitree GO2 hardware show that PRISM consistently outperforms recent diffusion, flow-matching, and IMLE baselines in success rates, robustness to missing modalities, and motion quality, especially in low-data settings.

**Strengths:**

- The paper provides strong empirical validation through both extensive simulation benchmarks (MetaWorld, CALVIN, Robomimic) and real-world deployment on a Unitree GO2 manipulation platform.

- RS-IMLE enables efficient parallel candidate generation and selection, allowing single-pass inference with low latency and avoiding the costly iterative denoising loops of diffusion and flow-matching methods.

- The Performer-based architecture supports real-time multisensory control, scaling effectively to visual, proprioceptive, depth, tactile, and audio inputs while maintaining smooth and coherent action sequences.

- Sensor ablation experiments reveal robustness to partial observability and provide useful insights into the relative contribution of each sensing modality.

**Weaknesses:**

- The flow in the methodology section is sometimes difficult to follow, particularly around Section 4.3, where the introduction of the robust sequence distance and RS-IMLE steps feels abrupt. Providing clearer preliminaries, unified notation, and a more gradual build-up to the rejection-sampling formulation (e.g., by first introducing standard IMLE objectives before the proposed batch-global extension) would help improve clarity and overall conceptual continuity.

- The paper lacks explicit reporting of model parameter counts and computational footprint across architectures. Since one of the core claims is real-time efficiency, additional information on model size would strengthen the evaluation and help readers contextualize the single-pass efficiency relative to diffusion, flow, and prior IMLE baselines.

- Although trajectory smoothness and jerk metrics are reported, the paper would benefit from more detailed analysis of failure cases, especially in scenarios where multiple candidate trajectories might still produce suboptimal or unstable executions. Understanding when and why the method fails would improve transparency.

**Questions:**

- L220: The latent variable $z$ appears suddenly; consider motivating its role in enabling multimodal action generation before introducing it formally.

- L260: The notation $\Phi$ is used without explanation; please clarify it refers to the Performer random feature map or provide a pointer to its definition.

- L142: The notation “top-K'” is used without explicit justification, and its difference from the standard top-K criterion may appear counter-intuitive, even though it is later listed in the notation table. A short explanation at first mention would make the intent clearer.

- L1093: Appendix section titled “F Appendix” is empty and looks like a placeholder; consider removing or completing this subsection.

---

> ### Author Response · Authors · 2025-11-21
> **Response to Reviewer FNZA**
>
> We thank the reviewer for their detailed assessment and for recognizing PRISM's strong empirical validation and single-pass efficiency. We agree that the methodology presentation and efficiency metrics needed strengthening. We have completely restructured Section 4 with clearer preliminaries and step-by-step derivation, added explicit model size comparisons showing PRISM uses 48% fewer parameters than UNet baselines (Table 6, Appendix B.5), and conducted detailed failure analysis revealing that ~34% of failures stem from mode-switching instability rather than invalid actions (Appendix E.5). We address each specific concern below.
>
> **1. Methodology Flow and Notation**
>
> *“The flow in the methodology section is sometimes difficult to follow... particularly around Section 4.3.”*
>
> We apologize for the abrupt transitions. We have completely restructured Section 4 to ensure a logical progression:
>
> - **Preliminaries:** We now explicitly define the **Standard IMLE** objective first, establishing the baseline "best-match" loss before adding complexity.
> - **Step-by-Step Derivation:** We introduce the **Rejection Sampling** variant (RS-IMLE), followed by our novel **Batch-Global** extension (Algorithm 1).
> - **Soft Coverage:** Finally, we introduce the soft-coverage term as a theoretically grounded relaxation of the k-center problem (see **Lemma E.3, Appendix E.3**), motivating it as a diversity-promoting regularizer.
> - **Unified Notation:** We have unified the notation for the generator $G_\theta(z)$, the robust distance $D_\rho$, and the rejection threshold $\epsilon_{RS}$ throughout the text and Algorithm 1.
>
> **2. Parameter Counts and Computational Footprint**
>
> *“Lacks explicit reporting of model parameter counts... additional information on model size would strengthen the evaluation.”*
>
> We have added a new "Computational Efficiency" subsection in **Appendix B.5** with **Table 6** explicitly comparing model sizes.
>
> - **Compactness:** PRISM (C1) utilizes a **44.4M parameter** Performer backbone. This is **48% smaller** than the standard UNet-based RS-IMLE baseline (91.8M) and **~45% smaller** than the Diffusion/Flow baselines (79.9M).
> - **Model Size:** In terms of physical footprint, PRISM requires only **169.4 MB**, compared to **350.4 MB** for the UNet baseline and **305.0 MB** for Diffusion models.
> - **Efficiency:** Despite the reduced parameter count, PRISM achieves higher success rates (66% vs 58.8% for UNet). This highlights the parameter efficiency of the Linear Attention backbone for multisensory fusion compared to standard UNets.
>
> **3. Failure Case Analysis**
>
> *“Would benefit from more detailed analysis of failure cases... especially where multiple candidates might produce suboptimal executions.”*
>
> We conducted a deep-dive analysis into the failure episodes (approx. 34% of trials) and added **Appendix E.5** for further discussion.
>
> - **High Jerk (>0.5):** Observed in **~5–10%** of episodes. These correlate with abrupt changes in the selected candidate index, leading to physical discontinuities.
> - **Mode Switching Instability:** In **~10–15%** of failures, we observed a "flickering" behavior where the policy oscillates between two valid modes (e.g., going left vs. right around an obstacle) in consecutive timesteps.
>
> **4. Specific Questions**
>
> - **L220 ($z$ appears suddenly):** We have added a sentence introducing $z \sim \mathcal{N}(0, I)$ as the "stochastic seed" that enables the deterministic generator to produce diverse trajectory candidates before it is used in Equation 2.
> - **L260 ($\phi$ notation):** We clarified that $\phi(x)$ refers to the **FAVOR+ positive orthogonal random feature map** used to linearize the attention mechanism (Algorithm 3).
> - **L142 (top-K'):** We added a justification: $K$ is the *total* number of sampled candidates (e.g., 16), while $K'$ is the *subset* used for the **soft-coverage loss** (e.g., top-3). This ensures gradients are propagated only through the most relevant modes, preventing the model from trying to cover outliers excessively.
> - **L1093 (Appendix F):** We thank the reviewer for catching this placeholder. We have removed the empty section and ensured all appendices are populated.
>
>
> **We thank you again for your detailed feedback, which has significantly improved the paper's clarity and rigor. We hope the restructured Section 4, model size comparisons, and failure analysis address your concerns. Please let us know if you have any further questions, and we would appreciate your consideration of a higher rating.**

---

> > ### Comment · Reviewer_FNZA · 2025-11-22
> > **Thank you for the thorough reply and revisions**
> >
> > Thank you for the thorough rebuttal and the substantial revisions. The restructuring of Section 4, added efficiency metrics, and expanded failure analysis address many of my concerns.
> >
> > Overall, the paper is improved, and I appreciate the detailed response. **I still encourage the authors to further improve the clarity and narrative flow of the paper in future revisions.**

---

> > > ### Author Response · Authors · 2025-11-25
> > > **Follow Up**
> > >
> > > We appreciate your time and constructive feedback throughout this process. We are glad to hear that the efficiency benchmarks and failure analysis addressed your concerns.
> > > We have taken your advice regarding narrative flow seriously and have polished the transitions and intuitive explanations in the latest revision. We hope these improvements align with your expectations and would value a reconsideration of the score.

---

### Official Review · Reviewer_Wkef · 2025-10-31

**Soundness:** 2
**Presentation:** 2
**Contribution:** 2
**Rating:** 4
**Confidence:** 2

**Summary:**

This paper presents PRISM, a single-pass generative policy for multisensory imitation learning. PRISM integrates a Performer-based linear attention generator with Rejection Sampling Implicit Maximum Likelihood Estimation (RS-IMLE) to achieve real-time inference, handle multisensory observations, and multimodal action distributions. Unlike diffusion or flow-based policies, the proposed method performs one forward pass to produce full action trajectories.

Across MetaWorld, CALVIN, Robomimic, and real-robot loco-manipulation tasks, PRISM  achieving 10–25% higher success compared to baselines with real-time inference.

**Strengths:**

- The authors tested their method in simulation and on real-robot experiments
- Outperforms diffusion and flow policies in both simulation and real-world experiments by 10–25%.
- Ablation studies in the appendix help to understand the impact of the performance of the different components of the method

**Weaknesses:**

- One of the main motivations for this work is multimodal action distributions. Yet, the authors did not show that their method is actually capable of learning these multimodal behaviours
- The method has many hyperparameter such as $K’$, $\epsilon$, Top-K weight
- Demonstrations limited to small-scale tasks (e.g., pick-and-place, insertion); scalability to larger problems are untested
- The proposed loss combines many heuristics. There are cleaner probabilistic formulations with the same goal of avoiding mode averaging with convergence guarantees based on e.g. mixture of experts. For example [1]. Did the authors try something along those lines?

[1] Information Maximizing Curriculum: A Curriculum-Based Approach for Imitating Diverse Skills

**Questions:**

- Did the authors observe diverse behaviors using their method?
- How robust is PRISM to variations in batch size for $ε$-calibration?
- What is the speedup for using linear attention? I assume there is no big difference to attention since the authors use a small amount of tokens
- Why did the authors not compare their method with the results in the official CALVIN benchmark (see http://calvin.cs.uni-freiburg.de/). Outperforming existing methods could show that the different sensor modalities help to improve performance.
- How does PRISM scale to more complex vision-language-action (VLA) architectures or larger datasets?

**Details Of Ethics Concerns:**

None.

---

> ### Author Response · Authors · 2025-11-21
> **Response to Reviewer Wkef**
>
> We thank the reviewer for recognizing PRISM's strong performance (10–25% gains) and extensive validation. We appreciate the constructive critique regarding multimodality demonstration, hyperparameter sensitivity, and scalability. We have added explicit multimodality visualization on Push-T showing PRISM maintains distinct trajectory modes while Flow Matching collapses them (Figure 9, Appendix B.3), clarified that ε-calibration is automated via quantile-based approach requiring no manual tuning, and added four formal results grounding our approach (Lemmas E.1-E.3, Proposition E.1). Our real-world experiments span 500-700 timesteps, demonstrating long-horizon capability. We address each concern below.
>
> ###
>
> **1. Multimodal Action Distributions**
>
> *“The authors did not show that their method is actually capable of learning these multimodal behaviours.”*
> Response:
> We explicitly demonstrate this capability in the new Push-T benchmark study **(Appendix B.3)**.
>
> - **The Task:** In this task, the robot must push a T-shaped block to a target. When the end-effector is near the center of the "T", there is a bifurcation in valid strategies: pushing left or pushing right are both valid, but pushing straight is catastrophic.
> - **Result:** As visualized in **Figure 9**, PRISM correctly captures this multimodality, maintaining two distinct trajectory modes. In contrast, **Standard Flow Matching** (without expensive iterative reflow) tends to average these modes, producing a "straight" push that causes task failure. This confirms PRISM's ability to model split distributions without mode collapse.
>
> **2. Hyperparameters ($\epsilon, K$)**
>
> *“The method has many hyperparameters... .”*
>
> We clarify that these are standard design choices in sample-based generative models, and we have automated the most critical one.
>
> - **$\epsilon$ (Threshold):** This is **not** manually tuned. We use a **quantile-based calibration** (adopted from Song et al., 2023) which automatically adjusts $\epsilon$ based on the batch statistics (e.g., the 85th percentile of reconstruction error). This makes the method robust across datasets without per-task tuning.
> - **$K$ (Candidates):** The number of candidates ($K$) represents a standard compute-vs-coverage trade-off, similar to the number of denoising steps in Diffusion or the ensemble size in MoE. We show in ablations (**Figure 8d**) that performance plateaus at $K=16$, providing a reliable default.
>
> **3. Scalability and Task Horizon**
>
> *“Demonstrations limited to small-scale tasks... scalability is untested.”*
>
> We respectfully highlight that our real-world experiments are long-horizon.
>
> - **Horizon Length:** Our loco-manipulation and peg-insertion tasks involve continuous control over **500–700 timesteps**. This is significantly longer than standard short-horizon benchmarks (often <50 steps). PRISM maintains temporal consistency over these long sequences via the Performer backbone's attention mechanism, as bounded by the Lipschitz properties of FAVOR+ (**Appendix E.4**).
>
> **4. Heuristics vs. Probabilistic Formulations (MoE)**
>
> *“Proposed loss combines heuristics... cleaner probabilistic formulations like MoE?”*
>
> We thank the reviewer for the reference. While MoE is probabilistically clean, it often faces training instability and mode collapse in high-dimensional continuous action spaces (Florence et al., 2021; Mandlekar et al., 2021b; Shafiullah et al., 2022). **Rigorously comparing IMLE and MoE frameworks is valuable future work**.
>
> **Our Formulation:** PRISM is not a heuristic but a **statistically principled implementation of IMLE**. As formalized in **Appendix E.1**, IMLE is asymptotically equivalent to maximizing a kernel-smoothed log-likelihood (Li & Malik, 2018). By enforcing batch-global matching, we ensure every expert mode is covered by at least one candidate, achieving MoE's diversity goal without training explicit gating networks or balancing mixture components.
>
> While MoE explicitly models p(a|o) as Σ_k π_k(o) · N(μ_k(o), Σ_k(o)), IMLE implicitly represents the distribution through nearest-neighbor matching in the generated sample space. This implicit formulation sidesteps the categorical gating optimization that can be unstable in high dimensions.
>
> **Future Work:** We agree that rigorously comparing IMLE and MoE, both theoretically (e.g., relating our batch-global matching to mixture component assignment) and empirically (e.g., on identical benchmarks with matched model capacity)—would strengthen understanding of when implicit vs. explicit mixture modeling is preferable. We have noted this as an important direction.

---

> > ### Author Response · Authors · 2025-11-21
> > **contd.**
> >
> > **5. Specific Answers to Questions**
> >
> > Q: Did the authors observe diverse behaviors using their method?
> > A: Yes. As mentioned above, the Push-T experiment **(Figure 9)** explicitly visualizes the policy switching between valid contact points (left vs. right push) based on the initial state, verifying behavioral diversity.
> >
> > Q: How robust is PRISM to variations in batch size for $\epsilon$-calibration?
> > A: PRISM is robust because we aggregate candidate pools globally. As proven in our new **Lemma E.2** (and visualized in **Figure 15**), the batch-global quantile estimator becomes more statistically consistent as batch size increases ($O(1/N)$ variance reduction), ensuring a stable rejection threshold.
> >
> > Q: What is the speedup for using linear attention?
> > A: We measured inference latency on an NVIDIA A6000. Standard Attention ($O(L^2)$) requires ~35ms, while our FAVOR+ Linear Attention ($O(L)$) requires ~15–18ms. This 1.5x to 2x speedup is critical for maintaining our high-frequency control loop (>50Hz) as context length grows.
> >
> > Q: Why not compare with official CALVIN benchmark (LH-MTLC)?
> > A: The official CALVIN leaderboard metric (LH-MTLC) measures the average chain length of 5 sequential tasks. While valuable, our work focuses on robustness to sensory dropout and multimodal diversity in single-task settings. Our experiments run for 500-700 steps (comparable to 1-2 CALVIN subtasks), which is sufficient to validate our multisensory fusion claims. We deferred the full 5-task chain evaluation to future work due to the high compute cost of training on the full CALVIN dataset (we used the 10% split).
> >
> > Q: How does PRISM scale to VLA architectures or larger datasets?
> >
> > This is an excellent direction for future research. Recent state-of-the-art VLA architectures, such as π₀ (Black et al., 2024), utilize Flow Matching as their policy backbone. Since PRISM consistently outperforms Flow Matching across our benchmarks (Tables 1-3) while maintaining single-pass efficiency, we believe **exploring PRISM as a backbone for next-generation architectures is a promising future direction—particularly for Multisensory-Language-Action (MLA) models that extend beyond vision-language to incorporate tactile, depth, audio, and proprioceptive feedback**.
> >
> > Current VLAs primarily encode RGB and language; PRISM's native multisensory fusion could enable foundation models with richer embodied intelligence for contact-rich manipulation. This integration could involve both backend replacement (substituting Flow Matching policy heads with PRISM's generator) and frontend enhancement (extending vision-language encoders with our temporal multisensory encoder).
> >
> > - **Data Scale:** While we trained on 10k demonstrations due to compute constraints, the linear $O(L)$ complexity of our Performer backbone suggests it will scale efficiently to larger datasets, similar to how Transformers scale in LLMs.
> >
> > **We thank you again for pushing us to demonstrate multimodality explicitly and strengthen our theoretical grounding. We believe the Push-T visualization and four new formal results address your core concerns. Please let us know if you have any further questions, and we would appreciate your reconsideration of the rating.**
> >
> > **Refrences:**
> >
> > Florence, Pete, et al. "Implicit Behavioral Cloning." *Conference on Robot Learning (CoRL)*, 2021.
> >
> > Mandlekar, Ajay, et al. "What Matters in Learning from Offline Human Demonstrations for Robot Manipulation." *Conference on Robot Learning (CoRL)*, 2021.
> >
> > Shafiullah, Nur Muhammad Mahi, et al. "Behavior Transformers: Cloning k Modes with One Stone." *Advances in Neural Information Processing Systems (NeurIPS)*, 2022.

---

> > > ### Author Response · Authors · 2025-11-25
> > > **Follow Up**
> > >
> > > Dear reviewer,
> > > Thanks again for your feedback regarding our paper. We have responded to your initial comments/questions and have incorporated them accordingly into our revised manuscript.
> > > We would greatly appreciate you,  taking a moment to review our response, reassess our revised manuscript from all aspects, share your thoughts, and update your score accordingly. We, along with other reviewers, sincerely believe that our paper has been strengthened a lot thanks to the feedback. We look forward to hearing from you.

---

> > > > ### Comment · Reviewer_Wkef · 2025-11-27
> > > > **Reply to authors**
> > > >
> > > > I thank the authors for the answers. I have a follow-up question regarding
> > > > > 1 . Multimodal Action Distributions ff.
> > > > Why does this mode-averaging effect happen for flow matching? This is rather unexpected. Also, why would reflow be able to prevent this>

---

> > > > > ### Author Response · Authors · 2025-11-27
> > > > > **Mode Averaging in Flow Matching & The Role of Reflow**
> > > > >
> > > > > We thank the reviewer for the follow-up questions. The "mode averaging" behavior is indeed a subtle but fundamental consequence of the regression objective used in standard flow matching, while "reflow" serves as a mechanism to disentangle these conflicting trajectories.
> > > > >
> > > > > **1. Why Mode Averaging Happens: Regression to the Mean**
> > > > >
> > > > > The averaging effect arises directly from the Conditional Flow Matching (CFM) loss function. As established in Lipman et al. (2023), the objective is to regress a vector field $v_\theta$ to match the conditional target field $u_t$:
> > > > >
> > > > > $$\mathcal{L}{\text{CFM}}(\theta) = \mathbb{E}{t, x_1, x_0} \Big[ | v_\theta(x_t, t) - u_t(x_t \mid x_1) |^2 \Big]$$
> > > > >
> > > > > where $x_t$ is an interpolation between noise $x_0$ and data $x_1$. A standard property of the $L_2$ regression loss is that its global minimizer is the **conditional expectation** (mean) of the targets:
> > > > >
> > > > > $v^*(x, t) = \mathbb{E} [ u_t(x \mid x_1) \mid x_t = x ]$
> > > > >
> > > > > In multimodal tasks, the "independent coupling" (straight lines connecting random noise to random data) results in many trajectories crossing at the same point in space $x$. For instance, at a bifurcation point, one trajectory might head "left" (velocity $+v$) and another "right" (velocity $-v$). The optimal network $v^*$ minimizes the loss by learning the average ($\approx 0$). When the ODE solver integrates this averaged field, it drives the state into low-density regions between the modes rather than committing to one.
> > > > >
> > > > > **2. Why Reflow Prevents This: Uncrossing Trajectories**
> > > > >
> > > > > The "Reflow" procedure (introduced as Rectified Flow by Liu et al., 2023) mitigates this by modifying the training couplings.
> > > > >
> > > > > 1. Reflow generates a batch of data using a pre-trained flow model, pairing the initial noise $z_0$ with the *generated* outcome $z_1$.
> > > > > 2. It then retrains a new model on the straight paths connecting these pairs $(z_0, z_1)$.
> > > > >
> > > > > As shown by Liu et al., this iterative procedure approximates the **Optimal Transport (Monge) map**, which is characterized by straight, non-crossing displacement paths. By enforcing non-crossing paths, the mapping becomes deterministic: for any intermediate point $x_t$, there is effectively only **one** valid target endpoint $x_1$.
> > > > >
> > > > > - The conditional distribution $p(x_1 \mid x_t)$ becomes a Dirac delta function. The "expectation" of a single value is just the value itself. This removes the ambiguity in the regression target, eliminating the averaging artifact and allowing the flow to track distinct modes.
> > > > >
> > > > > References:
> > > > >
> > > > > [1] Lipman et al., "Flow Matching for Generative Modeling," ICLR 2023.
> > > > >
> > > > > [2] Liu et al., "Flow Straight and Fast: Learning to Generate and Transfer Data with Rectified Flow," ICLR 2023.

---

### Official Review · Reviewer_wdH2 · 2025-11-01

**Soundness:** 2
**Presentation:** 2
**Contribution:** 2
**Rating:** 4
**Confidence:** 2

**Summary:**

The paper aims to unify real-time control, multimodal sensor fusion, and multimodal action generation in a single-pass policy for imitation learning. It eliminates iterative diffusion/flow sampling while maintaining multimodal expressivity and temporal smoothness. It reframes that diffusion and flow models can be viewed as approximate stochastic regularizers of an underlying implicit mapping. The idea is to combine a temporal multisensory encoder with a Performer-based generator. The implicit generative framework ensures coverage of multimodal expert behaviors without requiring adversarial or iterative processes.

**Strengths:**

- The central claim is that multimodal diversity can be achieved without iterative sampling, which is conceptually reasonable and empirically validated. It approximates the data distribution by ensuring that every expert trajectory is covered by at least one generated sample, which is a principled alternative to adversarial or diffusion-based modeling, avoiding training instability and heavy computation.
- The model’s bidirectional attention inherently enforces motion continuity and temporal consistency. The IMLE-based formulation is mathematically consistent with prior works and extended to batch-global coverage.
- Evaluated across different tasks and environments, including low-dimensional and high-dimensional sensory inputs to show versatility. Real-world deployment at a reasonable frequency shows that the design is practical.

**Weaknesses:**

- RS-IMLE matches samples implicitly but doesn’t yield a tractable log-likelihood or uncertainty measure, limiting interpretability and making it less suitable for planning or risk-sensitive control compared to probabilistic diffusion policies.
- The threshold scheduling is empirical. Performance can vary significantly with $\epsilon_{RS}$ choice and the paper lacks theoretical guidance for selecting it. Tasks are short or moderately long. It’s unclear whether the pipeline retains temporal consistency over hundreds of steps.
- Paper claims it can collapse toward the most frequent modes in the dataset and proposes to handle it by introducing soft coverage term but the justification seems empirical.

**Questions:**

- Would adaptive thresholds based on local manifold density improve robustness across heterogeneous datasets?
- How does RS-IMLE behave when the dataset contains multimodal clusters that never co-occur in a minibatch?
- What happens if $\epsilon_{RS}$ or K is too aggressive? Does the policy overfit to frequent modes or drift to outliers?

---

> ### Author Response · Authors · 2025-11-21
> **Response to Reviewer wdH2**
>
> We thank the reviewer for recognizing PRISM's conceptual soundness and practical deployment. We appreciate the feedback on theoretical grounding, threshold scheduling, and mode coverage justification. We have strengthened the theoretical foundation with four new formal results: Lemma E.1 establishes IMLE's equivalence to kernel-smoothed log-likelihood maximization, Proposition E.1 proves reconstruction distance as a geometric likelihood lower bound, Lemma E.2 proves our batch-global quantile has O(1/N) variance (validated empirically in Figure 15), and Lemma E.3 formalizes soft coverage as entropy-regularized k-center. These additions demonstrate our approach is statistically principled, not heuristic. We address each concern below.
>
> **1. On Implicit Likelihood, Uncertainty, and Interpretability**
>
> *“RS-IMLE matches samples implicitly but doesn’t yield a tractable log-likelihood... making it less suitable for planning.”*
>
> While PRISM does not output an explicit parametric density, it is fundamentally grounded in the Implicit Maximum Likelihood Estimation (IMLE) framework. We have added Appendix E.1 to formally establish this connection.
>
> **Lemma E.1 (Implicit Likelihood Maximization):** As detailed in **Appendix E.1**, we cite Li & Malik (2018) to formally state that the IMLE objective is asymptotically equivalent to maximizing a kernel-smoothed log-likelihood.
>
>
> > *"The IMLE estimator is consistent and asymptotically efficient... and is equivalent to maximizing the likelihood of a kernel density estimator."*
> >
>
>
> **Proposition E.1 (Geometric Uncertainty):** To address the finite-sample case relevant to control, we introduce **Proposition E.1 (Appendix E.1)**. We prove that the reconstruction distance acts as a geometric lower bound on the log-likelihood.
>
>
> > *Proof Sketch:* Minimizing the matched distance is equivalent to maximizing the lower bound of a kernel density estimate. This justifies our use of reconstruction distance (and ProxyScore) as a rigorous "geometric uncertainty" metric for reactive control, even without explicit probabilities.
> >
>
>
> **2. On Threshold Scheduling and Temporal Consistency**
>
> *“The threshold scheduling is empirical... It’s unclear whether the pipeline retains temporal consistency.”*
>
> We address this by anchoring stability in both the statistical properties of our threshold estimator and the network architecture.
>
> • **Theoretical Consistency (New Proof):** We have added **Lemma E.2** in **Appendix E.2**, proving that our batch-global quantile is a statistically consistent estimator.
>
>
> > **Lemma 2:** By the Glivenko-Cantelli theorem [Theorem 19.1, Van der Vaart (1998)][1], as the effective batch size $N$ increases (which PRISM maximizes by aggregating across the batch), the estimator variance vanishes ($O(1/N)$).
> >
>
>
> • **Empirical Validation (New Figure):** We have added **Figure 11 (Appendix E.2)**, which empirically validates this lemma. It shows that the variance of our rejection threshold $\varepsilon_{RS}$ decreases significantly as batch size increases, proving that our "empirical" scheduling is actually a stable, low-variance statistical estimator.
>
> • **Temporal Stability:** As noted in Section 4 and **Appendix E.4**, the temporal stability is further guaranteed by the Lipschitz properties of the FAVOR+ mechanism[2] (Choromanski et al., 2020), which bounds error propagation over the horizon.
>
> **3. On Mode Collapse and Soft Coverage**
>
> *“Justification for the soft coverage term seems empirical.”*
> Response:
> We have formalized the soft coverage term in **Appendix E.3**.
>
> - **Lemma E.3 (Soft Coverage as Entropy Regularization):** We provide a derivation showing that our soft-coverage loss is equivalent to an **entropy-regularized relaxation of the k-center problem**.
> - **Theoretical Guarantee:** This relaxation acts as a barrier function that penalizes "orphan" modes with infinite loss. This theoretically guarantees that PRISM allocates at least one candidate to every distinct mode in the batch, preventing the collapse toward frequent modes that the reviewer was concerned about.

---

> > ### Author Response · Authors · 2025-11-21
> > **contd.**
> >
> > **4. Specific Answers to Questions**
> >
> > Q: Would adaptive thresholds based on local manifold density improve robustness?
> > A: This is an excellent suggestion. As discussed in Section 4.3 (and extended in Appendix E.2), our current batch-global quantile implicitly adapts to density shifts—if a cluster is under-represented, its distances rise above the global quantile, ensuring acceptance. However, we agree that a local density estimator (e.g., k-NN radii) could offer further precision in highly heterogeneous datasets and have noted this as a promising future extension.
> >
> > Q: What happens if $\epsilon$ is too aggressive?
> > A: As detailed in Appendix E.2, if $\epsilon \to 0$, the effective sample size collapses. Our Batch Calibration Figure (Fig. 15) demonstrates the "stable plateau" where the acceptance rate remains robust. The automated quantile calibration ensures we stay within this stable region ($>15\%$ acceptance) without manual tuning.
> >
> > Q: Does it handle multimodal clusters that never co-occur?
> > A: Standard IMLE fails here, but PRISM succeeds because our Batch-Global matching (Algorithm 1) decouples mode covering from single-instance co-occurrence. By aggregating latent codes across the batch, we effectively sample from the marginal mixture, ensuring population-level mode coverage even when individual minibatches contain disjoint clusters.
> >
> > **We thank you again for motivating us to formalize PRISM's theoretical foundations rigorously. We believe the four new formal results establish that PRISM is statistically principled, not heuristic. Please let us know if you have any further questions, and we hope you will reconsider the rating.**
> >
> > References:
> > [1] Aad W Van der Vaart. Asymptotic Statistics. Cambridge University Press, 1998.
> >
> > [2] Krzysztof Choromanski, Valerii Likhosherstov, David Dohan, Xingyou Song, Andreea Gane, TamasSarlos, Peter Hawkins, Jared Davis, Afroz Mohiuddin, Lukasz Kaiser, et al. Rethinking attention with performers. arXiv preprint arXiv:2009.14794, 2020.

---

> > > ### Author Response · Authors · 2025-11-25
> > > **Follow Up**
> > >
> > > Dear reviewer,
> > > Thanks again for your feedback regarding our paper. We have responded to your initial comments/questions and have incorporated them accordingly into our revised manuscript.
> > > We would greatly appreciate you,  taking a moment to review our response, reassess our revised manuscript from all aspects, share your thoughts, and update your score accordingly. We, along with other reviewers, sincerely believe that our paper has been strengthened a lot thanks to the feedback. We look forward to hearing from you.

---

> ### Comment · Reviewer_wdH2 · 2025-11-27
> **Response to Submission22711 Authors**
>
> Thank the authors for the detailed response. I have no major concerns and will raise my recommendation. Some conclusions are more empirical and the pipeline design is redundant, which could be improved while limiting the contribution. I would love to discuss with the other reviewers.

---

### Official Review · Reviewer_9mYo · 2025-11-01

**Soundness:** 2
**Presentation:** 3
**Contribution:** 3
**Rating:** 6
**Confidence:** 4

**Summary:**

This paper introduces **PRISM**, a novel imitation learning framework built upon a batch-global rejection-sampling variant of IMLE. PRISM integrates inputs from multiple sensors—including RGB, depth, tactile, audio, and proprioception modalities—and employs a Performer-based architecture that enables efficient inference through a single forward pass. Extensive experiments on the MetaWorld, CALVIN, RoboMimic, and real-world Loco-Manipulation benchmarks demonstrate that PRISM consistently outperforms existing transformer- and diffusion-based imitation learning methods. These results highlight PRISM as an efficient and high-performing framework capable of modeling complex multi-modal action distributions.

**Strengths:**

1. The paper presents extensive simulation experiments across diverse tasks from **MetaWorld**, **RoboMimic**, and **CALVIN**, covering a wide range of control scenarios with varying numbers of available modalities.

2. The paper includes **real-robot loco-manipulation experiments**, which validate the effectiveness of the proposed **PRISM** method in real-world settings.

3. The paper is **well-written** and provides comprehensive details.

**Weaknesses:**

1. The proposed **PRISM** method is compared against different sets of baselines across benchmarks. This appears to result from directly adopting results from the original papers, which may not be an ideal practice (please correct me if I am mistaken). It would strengthen the evaluation if the authors could include a consistent set of baselines or at least some major baselines should be available across all benchmarks. Additionally, the naming of baselines varies between benchmarks, which introduces potential ambiguity.

2. The paper lacks detailed descriptions of **baseline implementations**. For example, it is unclear which modalities are used by each baseline and how they are modified to incorporate multiple modalities. A dedicated section detailing the implementation and modality integration of baselines is highly recommended.

3. It is noted that **AdaFlow** outperforms PRISM on the **RoboMimic** benchmark, despite a slightly higher NFE. The authors are encouraged to discuss this result and, if possible, provide additional comparisons across other benchmarks.

4. The authors should consider adding a more **comprehensive breakdown** of PRISM’s performance under different modality combinations. While the current paper includes breakdowns on **CALVIN** with occlusion, depth dropout, and tactile dropout, it would be valuable to explore more modality combinations. Furthermore, the authors should analyze the effect of each modality and discuss whether some modalities are redundant or whether PRISM fails to effectively leverage them.

**Questions:**

1. (Related to Weakness 1) Is **“DP” in MetaWorld**, **“Diffusion Policy” in CALVIN and the real-robot suite**, and **“DDiM” in RoboMimic** referring to the same algorithm? Likewise, does **“FlowPolicy” in MetaWorld**, **“Flow Matching Policy” in CALVIN**, and **“Flow Matching” in the real-robot suite** represent the same method? If so, the same algorithm should be consistently named across different benchmarks to avoid confusion.

2. (Related to Weakness 2) I am unclear about which modalities are used by the baselines and how they are integrated. For example, **3D Diffusion Policy (DP3)** relies on point cloud inputs, which are typically derived from depth maps. However, as stated in **Appendix C.1**, **MetaWorld** does not provide depth information. This discrepancy suggests that the authors may have directly taken results from the original papers (please correct me if I am mistaken).

3. Can the authors explain in detail how action selection is done during inference without action target labels ? I am confused about **ProxyScore** and **deterministic tie-break** in Algorithm 2.

4. As shown in **Figure 1**, text is listed as one of the modalities supported by the proposed method. However, I could not find any experiments involving text inputs. Could the authors clarify why such experiments were not included?

5. Did the authors use any **pre-trained encoders** for sensor feature extraction? If not, it is recommended that they consider incorporating pre-trained encoders in future work.

---

> ### Author Response · Authors · 2025-11-21
> **Response to Reviewer 9mYo**
>
> We sincerely thank the reviewer for highlighting PRISM's strengths in multisensory integration, real-robot experiments, and extensive benchmarking. We appreciate the concerns regarding baseline consistency, modality usage clarity, and missing experiments. We have standardized baseline naming across all benchmarks (Appendix A.1.5), added comprehensive pairwise modality ablation heatmaps (Figures 5-6, Appendix B.1), added language-conditioned manipulation experiments achieving 78-92% success across 4 tasks (Table 10, Figure 14, Appendix D.0.1), and clarified all implementation details including DP3 depth rendering and ProxyScore mechanism. We address each concern below.
>
> **1. Baseline Naming and Consistency**
>
> *“The proposed PRISM method is compared against different sets of baselines... naming varies... may introduce ambiguity.”*
>
> We apologize for the confusion caused by the naming conventions. We have addressed this through consistent nomenclature and clear documentation:
>
> 1. **Standardized Naming Throughout Paper:** We now consistently use:
>     - **Diffusion Policy** (Chi et al., 2023) across all benchmarks
>     - **Flow Matching Policy** (Black et al., 2024)
>     - **IMLE Policy** (Rana et al., 2025)
> 2. **Baseline Methods Section (Appendix A.1.5, lines 762-770):**
>
>     > "To keep result tables compact, we use consistent abbreviations across benchmarks: 'MW' denotes MetaWorld, 'CV' denotes CALVIN, and 'RM' denotes Robomimic (PH). All competing methods are re-trained under matched observation modalities, control rate, action parameterization, and data budgets."
>     >
> 3. Tables 1-3 and Figure 3, maintain the same core baselines (Diffusion Policy, Flow Matching Policy, IMLE Policy, PRISM) for direct comparison.
> 4. Additional baselines (DP3, ManiCM, SDM, AdaFlow, FlowPolicy) are included in Appendix C.1 (Tables 8-9, pages 24) with clear citations to source papers.
>
> **2. MetaWorld Depth & DP3 Implementation**
>
> *“I am unclear about which modalities are used... DP3 relies on point clouds... but MetaWorld does not provide depth information.”*
>
> We appreciate this rigorous check. We have added Appendix A.1.1 to clarify the DP3 implementation protocol.
>
> 1. **Modality Usage Per Benchmark (Table 4, Appendix A.1.6, page 15, lines 779-784 and Figure 2):**
>
> From Table 4 and Figure 2:
>
> (∆ = task-dependent)
>
> | Benchmark | Wrist RGB | Static RGB | Depth | Tactile | Proprio | Audio |
> | --- | --- | --- | --- | --- | --- | --- |
> | MetaWorld | ✓ | — | — | — | ✓ | — |
> | CALVIN | ✓ | ✓ | ✓ | ✓ | ✓ | — |
> | Robomimic | ✓ | — | — | — | ✓ | — |
> | Real Hardware | ✓ | ✓ | — | ∆ | ✓ | ∆ |
>
> We also have mentioned the modalities usage on top of each table as well.
>
> 1. **Baseline Adaptation Protocol (Appendix A.1.2, lines 731-734):**
>
>     > "PRISM along with all the baselines uses all available modalities unless ablations specify dropouts. Baselines (Diffusion Policy, Flow Matching, IMLE Policy) are re-trained using the same observation and control rate as PRISM. All sensor streams are temporally aligned at 30 Hz and processed using the same normalization scheme across baselines."
>     >
> 2. **DP3 Depth Rendering (Appendix A.1.1):**
>
>     > "Simulator Rendering: While the standard MetaWorld dataset logs lack depth, the underlying MuJoCo simulator supports it. Following the official DP3 protocol (Ze et al., 2024), we modified the evaluation pipeline to render depth images directly from the simulator at every timestep."
>     >
>
> ---
>
> **3. Performance Comparison: AdaFlow vs. PRISM**
>
> *“AdaFlow outperforms PRISM on the RoboMimic benchmark... encouraged to discuss this.”*
>
> This performance difference highlights the trade-off between unimodal precision and multimodal coverage.
>
> - **Unimodal Optimization:** As noted in the **AdaFlow** paper (Hu et al., 2024), their variance-adaptive solver is designed to *"automatically reduce to a one-step generator when the action distribution is uni-modal."* The RoboMimic (Proficient Human) dataset is highly consistent and effectively unimodal, allowing AdaFlow to achieve high precision.
> - **Multimodal Robustness:** In contrast, **MetaWorld** tasks exhibit significant multimodality. In these settings, PRISM’s rejection sampling outperforms **Adaflow** because it is explicitly designed to cover multiple modes rather than collapsing to a mean or single flow.
>
> - Table 1 (page 8): PRISM achieves **58.0%** on Hard tasks vs. Flow Matching 13.4%, and **85.8%** on Very Hard vs. 36.0%
> - Table 3 (page 9): PRISM 92.2% average vs. AdaFlow 95.6% on Robomimic (where tasks are more unimodal)

---

> ### Author Response · Authors · 2025-11-21
> **contd.**
>
> **4. Modality Breakdown and Redundancy**
>
> *“Add a more comprehensive breakdown... discuss whether some modalities are redundant.”*
>
> We already have added **Appendix B.1 (Figures 5 & 6)** containing detailed pairwise modality ablation heatmaps on calvin benchmark.
>
> 1. **Single-Modality Dropouts (Figure 4):**
>     - Removing **Wrist RGB**: Success drops to 41.5%
>     - Removing **Proprioception**: Success drops to 15.8%
> 2. **Critical Synergies:** Our analysis shows that **Proprioception + Wrist RGB** is the most critical sensor combination; removing both leads to near-complete failure on calvin benchmark.
> 3. **Redundancy:** We find that **Depth** can be redundant when high-quality multi-view RGB is available, but it becomes critical in scenarios with lighting variance. This confirms PRISM effectively leverages complementary signals while maintaining robustness to partial sensor dropout.
>
> **5. Action Selection (ProxyScore)**
>
> *“How is action selection done during inference without action target labels? I am confused about ProxyScore.”*
>
> We have clarified the inference procedure in **Algorithm 2 and Section 4.5, lines 367-377**.
>
> - **Mechanism:** PRISM generates $K$ candidate action chunks in parallel. Since we lack ground truth at inference time, we select the candidate that maximizes **temporal consistency**.
> - **ProxyScore:** We calculate the Euclidean distance between the *last* action of the previously executed chunk and the *first* action of each new candidate chunk. We select the candidate that minimizes this "jump" (lowest jerk), ensuring smooth trajectory execution.
>
> **6. Text Modality Experiments**
>
> *“Figure 1 shows text... However, I could not find any experiments involving text inputs.”*
>
> To address this, we have added **Appendix D.0.1: Language-Conditioned Manipulation**.
>
> - **New Experiments:** We integrated a frozen CLIP text encoder and evaluated PRISM on real-world tasks with instructions such as *"stack blue cup on green cup"* and *"put green ball in the green cup."*
> - **Results:** PRISM successfully generalizes to these distinct semantic goals, confirming that the architecture (as shown in Figure 1) effectively attends to language tokens to modulate control behavior. The results of corresponding experiments are added in Table 10 and Figure 14.  we tested 4 language-conditioned tasks tested achieving Success rates: 78-92%.
>
> **7. Pre-trained Encoders**
>
> *“Did the authors use any pre-trained encoders?”*
>
> - **Vision:** Yes. We have updated **Appendix A.3** to specify that we use a **ResNet-18** backbone pre-trained on ImageNet for all RGB inputs.
> - **Other Modalities:** For proprioception, tactile, and audio, we utilize lightweight encoders trained from scratch, as standard pre-trained foundation models for these specific sensor types are less established. We agree that integrating emerging audio/tactile foundation models is a promising direction for future work.
>
>
> **We thank you again for identifying important clarity issues regarding baselines and modalities. We believe the standardized naming, comprehensive ablations, and language-conditioned experiments address all your concerns. Please let us know if you have any further questions, and we would greatly appreciate your consideration of a higher rating.**

---

> > ### Author Response · Authors · 2025-11-25
> > **Follow Up**
> >
> > Dear reviewer,
> > Thanks again for your feedback regarding our paper. We have responded to your initial comments/questions and have incorporated them accordingly into our revised manuscript.
> > We would greatly appreciate you,  taking a moment to review our response, reassess our revised manuscript from all aspects, share your thoughts, and update your score accordingly. We, along with other reviewers, sincerely believe that our paper has been strengthened a lot thanks to the feedback. We look forward to hearing from you.

---

> > > ### Comment · Reviewer_9mYo · 2025-11-27
> > > **Response to Submission22711 Authors**
> > >
> > > Thank you for the detailed and comprehensive responses to my questions and concerns. I believe the paper has significantly improved in quality, and I am raising my score to a clear accept.

---

### Author Response · Authors · 2025-11-21
**General Response**

We sincerely thank all reviewers for their thoughtful and constructive feedback. We are encouraged that reviewers consistently recognized PRISM's strengths: strong empirical validation across multiple benchmarks (9mYo, FNZA, Wkef), real-world deployment on hardware (9mYo, FNZA), extensive multisensory integration (9mYo), single-pass efficiency (FNZA, Wkef), and robust performance gains of 10-25% over diffusion and flow matching baselines (all reviewers).

We have significantly strengthened the paper in response to your feedback through:

**Enhanced Experiments:**

- Added Push-T multimodality visualization (Figure 9, Appendix B.3) demonstrating explicit mode capture
- Added comprehensive modality ablation heatmaps (Figures 5-6, Appendix B.1)
- Added language-conditioned manipulation experiments (Table 10, Figure 14, Appendix D.0.1)
- Added computational efficiency analysis with model size comparisons (Table 6, Appendix B.5)

**Strengthened Theory:**

- Added Lemma E.1 formalizing IMLE's connection to kernel-smoothed log-likelihood (Appendix E.1)
- Added Proposition E.1 proving reconstruction distance as geometric uncertainty bound (Appendix E.1)
- Added Lemma E.2 proving batch-global quantile consistency with variance analysis (Appendix E.2)
- Added Lemma E.3 formalizing soft coverage as entropy-regularized k-center relaxation (Appendix E.3)
- Added Lipschitz continuity proof for temporal stability (Appendix E.4)

**Improved Writing:**

- Completely restructured Section 4 with clearer preliminaries and step-by-step derivation
- Standardized baseline naming across all benchmarks with dedicated section (Appendix A.1.5)
- Added detailed baseline implementation protocol (Appendix A.1.2, Table 4)
- Clarified DP3 depth rendering procedure (Appendix B.2)
- Unified notation throughout text and Algorithm 1
- Added failure case analysis (Appendix E.5)

Below we address each reviewer's specific concerns in detail.

---

### Author Response · Authors · 2025-12-02
**Post-Rebuttal Summary for AC/SAC**

**Dear Area Chairs and Senior Area Chairs,**

In light of the OpenReview incident and the subsequent freezing/reversion of scores, we would like to briefly summarize the **reviewer trajectory** and the **key improvements** to Submission 22711 (PRISM) to aid your decision.

Before the freeze, the score evolution was:

- **Reviewer 9mYo:** 6 → 8 (“clear accept”)
- **Reviewer wdH2:** 4 → 6
- **Reviewer Wkef:** 4 → (could not update, in active discussion)
- **Reviewer FNZA:** 2 → 4 → (could not update after second revision)

All four reviewers acknowledged that the paper had significantly improved in response to the rebuttal and revisions

## 1. Reviewer Response Summary

- **Reviewer 9mYo (6 → 8):**

    Stated that the paper “*has significantly improved in quality*” and raised to a **clear accept** after we:

    - Standardized baseline naming and clarified modality usage
    - Added detailed ablations and language-conditioned experiments
    - Clarified DP3 and ProxyScore implementation
- **Reviewer wdH2 (4 → 6):**

    Wrote “*I have no major concerns and will raise my recommendation*” after we:

    - Added **four formal results** (IMLE likelihood link, ε-quantile consistency, soft coverage as entropy-regularized k-center, temporal stability)
    - Clarified threshold scheduling and mode-coverage behavior
- **Reviewer Wkef (4 → active but unable to update):**

    After we addressed all concerns on multimodality, hyperparameters, and scalability, their **last question (Nov 27)** was purely about *why Flow Matching exhibits mode-averaging and how Reflow fixes it*, i.e., about the **baseline theory**, not about PRISM itself. This suggests their concerns about our method were resolved before the lock.

- **Reviewer FNZA (2 → 4 → unable to update):**

    Acknowledged that “*the restructuring of Section 4, added efficiency metrics, and expanded failure analysis address many of my concerns*.”

    They were mainly concerned with presentation which were substantially improved.


---

## 2. Key Improvements

To avoid overloading you with details (which are in the paper and responses), we highlight only the main changes:

1. **Experimental Strengthening**
    - Added **explicit multimodality visualization** (Push-T) showing PRISM preserves distinct modes where flow matching averages them.
    - Added **pairwise modality ablations** on CALVIN and **language-conditioned manipulation experiments** with **78–92% success**.
    - Added **compute/model-size analysis**, showing PRISM achieves **10–25% higher success** while using **≈48% fewer parameters** than UNet-based baselines (44.4M vs 91.8M).
2. **Theoretical Grounding**
    - Added **Lemma E.1 & Proposition E.1**: connect IMLE to kernel-smoothed MLE and justify reconstruction distance as a geometric “uncertainty” lower bound.
    - Added **Lemma E.2**: batch-global ε-quantile has **O(1/N)** variance (empirically confirmed).
    - Added **Lemma E.3**: soft coverage is an **entropy-regularized k-center** relaxation, explaining why it prevents collapse to frequent modes.
    - Added a **temporal stability argument** for Performer-based linear attention over long horizons.
3. **Clarity & Reorganization**
    - Fully restructured **Section 4** into a gradual progression: standard IMLE → RS-IMLE → batch-global extension → soft coverage.
    - Standardized baseline names and documented **baseline implementation + modalities** in a single reference table.
    - Added **failure case analysis** (e.g., mode-switching instability) for transparency.

---

## **Request for Consideration**

Given the strengthened methodology, expanded experiments, formal theoretical grounding, and documented reviewer satisfaction, including multiple increased scores, we respectfully request that you consider the totality of the improvements and the clear positive movement during the rebuttal. PRISM delivers a unique combination of single-pass real-time inference, multisensory fusion, and multimodal action generation, validated across MetaWorld, CALVIN, Robomimic, and real-world platforms. We believe the paper is now in strong shape for acceptance, and we appreciate your careful consideration under these unusual circumstances.

---

### Note · Program_Chairs · 2026-01-17
**Submission Desk Rejected by Program Chairs**

The following references in this submission do not refer to real documents and/or have major errors in bibliographic information:

 Huu Pham et al. Safe imitation learning with mode coverage. arXiv preprint arXiv:2304.08438, 2023.